# Nuts and Cardio-Metabolic Disease: A Review of Meta-Analyses

**DOI:** 10.3390/nu10121935

**Published:** 2018-12-06

**Authors:** Yoona Kim, Jennifer Keogh, Peter M. Clifton

**Affiliations:** 1Department of Food and Nutrition/Institute of Agriculture and Life Science, Gyeongsang National University, Jinju 52828, Korea; yoona.kim@gnu.ac.kr; 2School of Pharmacy and Medical Sciences, University of South Australia, General Post Office Box 2471, Adelaide, SA 5001, Australia; jennifer.keogh@unisa.edu.au

**Keywords:** nuts, meta-analyses, type 2 diabetes, cardiovascular disease

## Abstract

Objectives: Accumulating epidemiological and intervention evidence suggest that nut consumption is associated with reduced incidence of some cardiometabolic diseases. However, to date no review of meta-analyses of epidemiological and intervention studies has evaluated the effects of nut consumption on cardiometabolic disease. Design/Results: Electronic searches for meta-analyses of epidemiological and intervention studies were undertaken in PubMed^®^/MEDLINE^®^. Meta-analyses of prospective studies show that nut consumption appears to be associated with reduced all-cause mortality by 19–20% (*n* = 6), cardiovascular disease (CVD) incidence (19%; *n* = 3) and mortality (25%; *n* = 3), coronary heart disease (CHD) incidence (20–34%; *n* = 2) and mortality (27–30%; *n* = 2) and stroke incidence (10–11%; *n* = 7) and mortality (18%; *n* = 2). No association between nut consumption and the risk of type 2 diabetes mellitus (T2DM) was observed in meta-analyses of prospective studies, whereas a decrease in fasting blood glucose ranging from 0.08 to 0.15 mmol/L was observed in 3 meta-analyses of intervention studies. In the interventions, nut consumption also had favorable effects on total cholesterol (0.021 to 0.28 mmol/L reduction from 8 meta-analyses of interventions) and low-density lipoprotein cholesterol (0.017 to 0.26 mmol/L reduction from 8 meta-analyses of interventions) and endothelial function (0.79 to 1.03% increase in flow-mediated dilation from 4 meta-analyses of interventions). Nut consumption did not significantly affect body weight. Nut consumption had no effect on inflammatory markers in intervention studies. The effect on blood pressure was inconsistent. A higher nut consumption was associated with a lower incidence of hypertension in prospective studies, while nut consumption did not improve blood pressure in intervention studies. Conclusions: Nut consumption appeared to be associated with lower all-cause mortality and CVD and CHD mortality. There was no association between nut consumption and the incidence of T2DM although fasting blood glucose is decreased in intervention studies. In intervention studies nuts lower total cholesterol and low-density lipoprotein cholesterol (LDL-C).

## 1. Introduction

Cardiometabolic disease is a combination of disorders such as excessive visceral fat, hypertension, atherogenic dyslipidemia and glucose intolesrance which together result in type 2 diabetes mellitus (T2DM) and cardiovascular disease (CVD) [1].

Nuts have been reported to reduce the risk of T2DM and CVD [2,3,4,5]. High levels of unsaturated fatty acids and low levels of saturated fatty acids and bioactive compounds (polyphenols, carotenoids, phytosterols, fiber and minerals) may contribute to cardio-protection through improvement of glycemic control, lipid profiles, weight, blood pressure, endothelial function, oxidative status, antioxidant and anti-inflammatory action [6,7].

Diabetes, a key cardiometabolic disease is a global health concern with an expected increase in prevalence from 8.4% in 2017 to 9.9% in 2045 [8]. People with T2DM have an increased risk of CVD and up to 80% of individuals with T2DM die from CVD [9]. For this reason, primary prevention of both T2DM and CVD is vital. A healthy diet is important for the prevention and treatment of cardiometabolic disease such as T2DM and CVD [10,11].

This review aims to provide an update on the impact of nut consumption on preventable cardiometabolic diseases by reviewing meta-analyses of epidemiological studies and intervention trials. The evidence for nuts influencing diabetes risk and glycemic control as well as for nuts reducing strokes and weight is very mixed so this review provides a comprehensive review of these controversial areas.

## 2. Methods

We examined meta-analyses of the effects of nut consumption on cardiometabolic diseases in humans. The literature search was performed on the PubMed^®^/MEDLINE^®^ (https://www.ncbi.nlm.nih.gov/pubmed/) database restricted to full articles in English up to 29 September 2018. The search terms included meta-analysis combined with nut(s) or tree nut(s) or almond(s) or Brazil nut(s) or cashew nut(s) or hazelnut(s) or macadamia(s) or peanut(s) or pistachio(s) or walnut(s). Reference lists of selected papers were also examined. The titles of articles were initially screened and then abstracts and full-texts of literature were reviewed for final study selection. Inclusion criteria were nuts as outlined above, meta-analysis, all-cause mortality, mortality, incidence, CVD, coronary heart disease (CHD), stroke, T2DM, hypertension, metabolic syndrome, obesity, adiposity, glycemic control, glucose, lipids, blood pressure, inflammatory markers, endothelial function, flow-mediated dilation. Meta-analyses of human studies were included and Non-English articles were excluded. Thirty-four meta-analyses that specifically addressed these inclusion criteria were selected and included in this review. A flow chart of the study screening and selection process is reported in Figure 1.

## 3. Results

### 3.1. Prospective Cohort Studies

All cohort mortality studies are summarized in Table 1. Forest plots for Table 1 are presented in Appendix A.

#### 3.1.1. All-Cause Mortality 

Meta-analyses [5,12,13,14,15,16] of prospective studies that have examined the effect of nut consumption on all-cause mortality have found a ~19% reduction in all-cause mortality comparing the highest to the lowest nut intakes. Three meta-analyses [13,14,16] of prospective studies have reported a ~21% reduction in all-cause mortality per one serving (28 g)/day increase. All meta-analyses show excellent consistency although the number of studies varied in each one.

In the meta-analysis conducted by Chen et al. 2017 [12], nut consumption was related to a reduced risk of all-cause mortality with a relative risk (RR) of 0.81 (95% confidence interval (CI) 0.78, 0.84; *p*-het = 0.20; I^2^ = 22%) for highest versus lowest consumption in 16 studies (13 publications) [17,18,19,20,21,22,23,24,25,26,27,28,29]. Two additional studies [30,31] were included for the dose–response analysis. For risk reduction per serve of nuts non-linear associations (*p* < 0.001 for heterogeneity (*p*-het)) with all-cause mortality were seen in 18 studies (15 publications) [17,18,19,20,21,22,23,24,25,26,27,28,29,30,31] (766,470 subjects with 81,034 deaths) with a RR of 0.96 (95% CI 0.94, 0.97; I^2^ = 72.3%) per serve of nuts per week. The reduction in mortality levelled off at ~3 servings/week (12 g/day) of nuts.

Aune et al. 2016 [14] examined 15 studies (15 publications) [17,18,19,20,21,22,23,24,25,26,27,28,29,31,32] (819,448 subjects with 85,870 deaths)] and showed a RR of 0.81 (95% CI 0.77, 0.85; *p*-het = 0.05; I^2^ = 41%) for highest versus lowest nut consumption. Compared with the meta-analysis by Chen et al. 2017 [12], this meta-analysis [14] added 2 more studies [31,32]. A study [32] which appears not to have nuts in the results was included as the authors [14] obtained nut intake information from the first author of the study [32]. Even though the 2 meta-analyses [12,14] had different numbers of studies, subjects and deaths, these showed the same RR of 0.81. Four studies (3 publications) [17,18,19] (20,2751 subjects with 42,508 deaths) showed a RR of 0.80 (95% CI 0.74, 0.86; *p*-het = 0.07; I^2^ = 58%) for tree nuts, while 5 studies (3 publications) [17,18,19] (265,252 subjects with 44,396 deaths) showed a RR of 0.85 (95% CI 0.82, 0.89; *p*-het = 0.30; I^2^ = 18%) for peanuts. In a dose-response analysis, a nonlinear association was observed (*p* < 0.0001) showing no further decrease in risk above 15–20 g/day of nut consumption. The RR per 28 g/day in 16 studies (15 publications) [17,18,19,20,21,22,23,24,25,26,27,28,29,31,32] was 0.78 (95% CI 0.72, 0.84; *p*-het < 0.0001; I^2^ = 66%).

Other meta-analyses were very similar [5,12,15,16]. Schwingshackl et al. 2017 [13] investigated the association between food groups and all-cause mortality in 16 prospective studies [17,18,19,21,22,23,24,25,26,27,28,29,30,31] with 80,204 deaths. This meta-analysis [13] excluded one study [20] as their analysis [13] excluded populations with chronic disease and studies containing only cause-specific mortality. One serving of nuts (28 g)/day increase was inversely associated with the risk of all-cause mortality (RR = 0.76; 95% CI 0.69, 0.84; I^2^ = 82%; *p*-het = 0.001) in a nonlinear dose-response manner. The RR was 0.80 (95% CI 0.74, 0.86; I^2^ = 84%; *p*-het < 0.001) when comparing the highest with the lowest nut consumption. 

Heterogeneity was higher in this analysis [13] but not in Chen et al. 2017 [12]. The higher heterogeneity is explained by between-study differences such as shorter-term follow-up and geographic location (with big differences between European and Asian studies) [13].

#### 3.1.2. Cause-Specific Mortality 

##### Diabetes Mortality 

Only one meta-analysis [14] examined the association between nut consumption and diabetes mortality showing a 32% reduction in diabetes deaths comparing the highest to the lowest nut intakes. It is difficult to know what the nature of a “diabetic death” was the International Classification of Diseases (ICD) code of 250 is very non-specific. The total number of deaths from diabetes were low in each paper. In addition, the study from Luu et al. 2015 was not significant in the original paper whereas it is shown as significant in the meta-analysis casting severe doubt on the conclusions of the meta-analysis.

Aune et al. 2016 [14] found an inverse association between nut consumption and deaths from diabetes in people with T2DM. For highest versus lowest nut consumption, the RR for total nuts from in the meta-analysis containing the studies from Luu et al. 2015 (Southern Community Cohort study and the Shanghai Health study), Bao et al. 2013 (The Nurses’ Health Study (NHS)), Bao et al. 2013 (HPFS) and van den Brandt et al. 2015 (Netherlands Cohort study) [17,18,19] (202,751 subjects with 800 deaths) was 0.68 (95% CI 0.52, 0.90; *p*-het = 0.59; I^2^ = 0%). The RR for peanuts [17,18,19] (265,252 subjects with 901 deaths) was 0.84 (95% CI 0.60, 1.19; *p*-het = 0.14; I^2^ = 42.6%) and the RR for tree [18,19] (130,987 subjects with 462 deaths) was 1.19 (95% CI 0.74, 1.89; *p*-het = 0.43; I^2^ = 0%).

##### CVD Mortality 

Three meta-analyses [5,12,15] of prospective studies reported a 19–25% lower rate of CVD mortality comparing highest to lowest consumption.

A recent meta-analysis of prospective studies by Chen et al. 2017 [12] indicated that nut consumption was associated with a lower risk of cardiovascular disease (CVD) mortality. An analysis of highest-versus-lowest consumption showed a 25% reduction (RR = 0.75; 95% CI 0.71, 0.79) in CVD mortality from 16 studies (13 publications) [17,18,19,20,21,22,23,25,26,27,28,29,33] (524,610 subjects with 19,574 deaths). Tree nuts and peanuts also showed similar reductions with RRs of 0.81 (95% CI 0.74, 0.89) and 0.78 (95% CI 0.73, 0.85) respectively. Earlier, smaller meta-analyses showed a 26–29% reduction in CVD in mortality [5,15].

##### Coronary Heart Disease Mortality

Chen et al. 2017 [12] found a 27% reduction (RR = 0.73; 95% CI 0.67, 0.80; *p*-het = 0.30; I^2^ = 14.2%) in coronary heart disease (CHD) mortality comparing the highest to the lowest intakes of total nuts from 13 studies (10 publications) [17,18,19,20,21,22,25,26,27,28] (429,833 subjects with 10,083 deaths). Tree nuts (21%) and peanuts (24%) also showed similar reductions. Nonlinear inverse associations between nut consumption and cause-specific mortality were found with the greatest decrease at 3 servings/week (12 g/day) (*p* < 0.001 for non-linearity), with no further additional reduction in mortality with increasing nut consumption [12]. A similar reduction was found in a smaller study [5].

##### Stroke Mortality

A meta-analysis including 12 prospective studies [12] showed an 18% reduction in stroke mortality while an earlier meta-analysis [5] which included only 3 studies showed a nonsignificant effect.

Chen et al. 2017 [12] showed a lower risk of stroke mortality after nut consumption with a RR of 0.82 (95% CI 0.73, 0.91; *p*-het = 0.82; I^2^ = 0%) from 12 studies (9 publications) [17,18,19,21,25,27,28,33,34] involving 449,293 subjects with 4398 deaths in a comparison of highest with lowest total nut consumption. On the other hand, a statistically nonsignificant reduction (RR = 0.83; 95% CI 0.69, 1.00) in stroke mortality was observed in analyses of both highest-versus-lowest consumption and dose-response conducted by Mayhew et al. 2016 [5] which only included 4 studies. 

#### 3.1.3. Incidence of Type 2 Diabetes, CVD, Hypertension, CHD and Stroke

A summary of nut consumption and incidence of T2DM, CVD, hypertension, CHD and stroke is shown in Table 2. Most meta-analyses were analyzed with prospective studies while two meta-analyses by Afshine et al. 2014 [57] and Weng et al. 2016 [58] were analyzed with prospective and intervention studies in Table 2. Forest plots for Table 2 are presented in Appendix A.

##### Type 2 Diabetes

Most meta-analyses have found no relationship with diabetes incidence but Wu et al. 2015 [59] has shown a 2% lower incidence. On the other hand, a dose response analysis meta-analysis [57] showed a 13% reduction in diabetes incidence per 4 servings/week.

Schwingshackl et al. 2017 [60] examined the association between food groups and incidence of T2DM in prospective studies with 27,016 occurrences of T2DM comparing the highest with the lowest nut consumption (average nut consumption from lowest to highest ntile (quartile or quintile) ranging from 0 to 27 g/day). They found no significant association between nut consumption and risk of T2DM with a RR of 0.95 (95% CI 0.85, 1.05; *p*-het = 0.004; I^2^ = 67%) in 8 prospective studies (6 publications) [53,54,55,56,61,62] while a 28 g/day increase did not significantly attenuate the risk of T2DM (RR = 0.89; 95% CI 0.71, 1.12; *p*-het = 0.0002; I^2^ = 77%) in 7 prospective studies (5 publications) [53,54,55,56,61]. However, given the marked heterogeneity this finding is not reliable. This finding was consistent with other meta-analyses [16,63,64]. If body mass index (BMI) is not adjusted for a relationship with T2DM can be seen. Luo et al. 2014 [16] showed a one serving/day increase in total nut consumption resulted in a RR of 0.88 (95% CI 0.84, 0.92; *p*-het = 0.60; I^2^ = 0.0%) after adjustment for age only (4 studies [53,54,55,56]). In a meta-analysis of 6 studies (5 publications) [53,55,56,65,66] (5 prospective cohorts and 1 randomized controlled trial (RCT; 230,216 subjects with 13,308 events) conducted by Afshin et al. 2014 [57], nut consumption per 4 servings/week decreased the risk of T2DM by 13% (RR = 0.87; 95% CI 0.81, 0.94; *p*-het = 0.269; I^2^ = 21.8%) [57]. Given that nuts are associated with a lower BMI adjustment for BMI may obscure the true relationship between nuts and T2DM which is directly mediated via lower BMI.

##### Cardiovascular Disease

Three meta-analyses [5,14,16] of prospective studies that examined high and low consumption of nuts have shown a lower incidence of CVD. Aune et al. 2016 [14] showed a 19% reduction while Mayhew et al. 2016 [5] and Luo et al. 2014 [16] showed a 44% and a 30% reduction, respectively. However, Mayhew et al. 2016 [5] and Luo et al. 2014 [16] just analyzed only one and four studies, respectively.

Aune et al. 2016 [14] found nut consumption was inversely associated with total CVD incidence with a RR of 0.81 (95% CI 0.74, 0.89; *p*-het = 0.02; I^2^ = 52.3%) from 12 cohort studies (12 publications) [17,18,19,21,22,25,27,28,29,31,35,36] comparing the highest with the lowest intakes. Consumption of tree nuts or peanuts showed a similar effect on CVD incidence with RRs of 0.81. The RR per 28 g/day increase in nuts from 12 cohort studies (12 publications) [17,18,19,21,22,25,27,28,29,31,35,36] was 0.79 (95% CI 0.70, 0.88). RRs were similar per 28 g/day increase in peanuts (RR = 0.64) or tree nuts (RR = 0.75) [14].

##### Hypertension

Meta-analyses of prospective studies [63,64,67] have found a ~15% lower risk of hypertension comparing the highest to the lowest nut intakes.

Schwingshackl et al. 2017 [67] examined the association between 12 food groups (whole grains, refined grains, vegetables, fruits, legumes, nuts, dairy, fish, eggs, red meat, processed meat and sugar-sweetened beverages) and incidence of hypertension in a meta-analysis of prospective studies. This meta-analysis of 4 studies [68,69,70,71] showed that 28 g/day of nuts attenuated the risk of hypertension by 30% (RR: 0.70; 95% CI 0.45, 1.08; I^2^ = 69%; *p*-het = 0.02). The RR was 0.85 (95% CI 0.78, 0.92; I^2^ = 0%; *p*-het = 0.92) in a comparison of high versus low nut consumption.

Guo et al. 2015 [63] found that nut consumption, especially at >2 servings/week was associated with a lower risk of hypertension. Comparing the highest with the lowest categories of nut consumption, a 16% decrease in the incidence of hypertension was observed (RR = 0.84; 95% CI: 0.76, 0.93) in 3 studies from 3 publications [68,70,72].

Zhou et al. 2014 [64] also found the highest nut consumption decreased the risk of hypertension by 15% (RR = 0.85; 95% CI 0.79, 0.92; *p*-het = 0.927; I^2^ = 0.0%) compared with the lowest nut consumption in an analysis of 4 studies from 4 publications [68,70,71,72] involving 40,102 subjects with 12,814 events.

##### Coronary Heart Disease

The incidence of CHD (or CAD) was lowered by 17–34% as shown in 7 meta-analyses [5,14,16,58,64,73,74] of prospective studies.

Aune et al. 2016 [14] found that nut consumption decreased the risk of CHD by 24% (RR = 0.76; 95% CI 0.69, 0.84) comparing the highest with the lowest intakes. This finding came from 11 studies from 10 publications [17,18,19,22,26,27,28,37,38,39] involving 315,397 subjects with 12,331 events. Consumption of either tree nuts or peanuts also showed similar reductions in CHD risk with RRs of 0.79 and 0.76 respectively. An increase in 1 serving/day resulted in a 21% decrease in CHD (RR = 0.79; 95% CI 0.70, 0.88) in 12 studies from 12 publications [17,18,19,21,22,25,27,28,29,31,35,36] (376,228 subjects with 18,655 events). Similar results were seen in earlier meta-analyses [5,16,58,64,74].

Bechthold et al. 2017 [73] showed a significant inverse association between nut consumption and risk of CHD (RR = 0.80; 95% CI 0.62, 1.03; I^2^ = 79%, *p*-het = 0.002) in 4 prospective cohort studies from 4 publications [37,38,39,40] with 5480 cases of CHD comparing groups with the highest and the lowest nut consumption (overall nut consumption ranging from 0 to 28 g/day). An additional intake of 28 g/day of nuts showed a 33% reduction in the risk of CHD (RR = 0.67; 95% CI 0.43, 1.05, I^2^ = 85%, *p*-het = 0.001). A non-linear dose-response was seen. An increase in nut consumption up to 10–15 g/day reduced the incidence of CHD by ~21%.

##### Fatal/Non-Fatal Coronary Heart Disease

A meta-analysis by Afshin et al. 2014 [57] showed that 4 serves/week of nuts lowered the risk of fatal CHD (RR = 0.76, 95% CI 0.69, 0.84; *p*-het = 0.227; I^2^ = 27.7%) and nonfatal CHD (RR = 0.78, 95% CI 0.67, 0.92; *p*-het = 0.463; I^2^ = 0.0%) by 24% and 22%, respectively. Per daily serve they found significant reductions in risk of non-fatal CHD and fatal CHD with RRs of 0.78 and 0.76 respectively. Non-fatal CHD was analyzed from 4 studies (4 publications) [2,38,39,47] involving 141,390 subjects with 2101 events, while fatal CHD events came from 6 studies (5 publications) [2,19,22,38,39] involving 206,114 subjects with 6749 events.

##### Stroke

All meta-analyses apart from two [14,75] which showed a 10–11% reduction in stroke incidence, have found no effect of nuts on stroke incidence. However, the meta-analysis of Aune et al. 2016 [14] is much more convincing as it included a greater number of studies compared with other meta-analyses [16,57,64,73].

Aune et al. 2016 [14] reported a 11% reduction (RR = 0.89; 95% CI 0.82, 0.97; *p*-het = 0.90; I^2^ = 0%) in the risk of stroke comparing the highest with the lowest nut consumption. This finding came from 10 studies (9 publications) [17,18,27,28,33,34,41,42,43] involving 396,768 subjects with 9272 events. One serve/day of reduced the risk of stroke by 7% (RR = 0.93; 95% CI 0.83, 1.05; *p*-het = 0.31; I^2^ = 13.7%) in 11 studies (9 publications) [17,18,27,28,33,34,41,42,43].

Shi et al. 2014 [75] also found an inverse association between nut consumption and risk of stroke in 4 prospective studies (3 publications) [34,42,43] including 228,799 subjects with 5669 strokes. Nut consumption reduced total stroke risk by 10% (RR = 0.90; 95% CI 0.81, 0.99; *p*-het = 0.527, I^2^ = 0) comparing the highest with the lowest nut consumption. Female nut consumers (RR = 0.85, 95% CI 0.75, 0.97) showed a lower incidence of stroke than male nut consumers (RR = 0.95, 95% CI 0.82,1.11).

However, Bechthold et al. 2017 [73] found no association between nut consumption and the risk of stroke (RR = 0.94; 95% CI 0.85, 1.05, I^2^ = 18%, *p*-het = 0.30) in 6 studies (6 publications) [33,41,42,43,44,76] with 7490 cases for stroke which compared the highest with the lowest nut consumption (ranging from 0–30 g/day). An increase of 28 g/day was not associated with stroke (RR = 0.99; 95% CI 0.84, 1.17, I^2^ = 45%, *p*-het = 0.11) in a non-linear dose-response. In addition, other meta-analyses found no association between nut consumption and risk of stroke [16,57,64]. In comparison with the meta-analysis by Aune et al. 2016 [14], these meta-analyses [16,57,64,73] included fewer studies.

#### 3.1.4. Body Weight

Li et al. 2018 [77] conducted a meta-analysis of prospective studies to examine the association between nut consumption and the risk of metabolic syndrome or overweight/obesity or obesity alone. The RR per one serving/week of nut consumption increase from 3 studies [78,79,80] was 0.96 (95% CI 0.92, 0.99; I^2^ = 0) for risk of metabolic syndrome. The RRs for overweight/obesity (0.97; 95% CI 0.95, 0.98; I^2^ = 0) and obesity (0.95; 95% CI 0.89, 1.02; I^2^ = 74%) were similar per one serving/week of nuts from 2 studies [81,82] and 2 studies [82,83], respectively.

Pooled results of the slope coefficient from a meta-analysis of epidemiological studies, which was performed by Grosso et al. 2015 [15], showed that nut consumption was inversely associated with BMI (0.15% (slope coefficient: −0.15; SE: −0.24, −0.06)) and smoking (−0.59% (slope coefficient: 0.59; SE −0.99, −0.20)). An additional one serving of nuts per week was associated with increased consumption of vegetables (~13 g/day (slope coefficient: 13.28; SE: 6.09, 20.46)), fruits (~10 g/day (slope coefficient: 9.82; SE: 4.5, 15.14)) and alcohol (~1 g/day (slope coefficient: 0.99; SE: 0.79, 1.20)) [15]. 

### 3.2. Weight, Blood Glucose, Lipids and Inflammatory Markers, Endothelial Function and Blood Pressure in Meta-Analyses of Intervention Studies 

#### 3.2.1. Body Weight

A summary of nut consumption and anthropometric parameters in RCTs is shown in Table 3. Forest plots for Table 3 are presented in Appendix A.

In a recent meta-analysis of RCTs by Guasch-Ferre et al. 2018 [94], a diet rich in walnuts did not alter weight [weighted mean difference (WMD) = −0.12 kg; 95% CI −2.12, 1.88; *p* = 0.90] and body mass index (BMI) (WMD = −0.11 kg/m^2^; 95% CI −1.15, 0.92; *p* = 0.82) compared with control diets in 10 studies from 10 publications [95,96,97,98,99,100,101,102,103,104] and 6 studies from 6 publications [95,96,97,105,106,107] respectively.

Perna et al. 2016 [108] conducted a meta-analysis of 3 studies from 3 publications [109,110,111] using a Bayesian random effect meta-analysis of mean differences from baseline across treatments (MDΔ) (e.g., a diet supplemented with hazelnuts vs. a control diet). They found that BMI [pooled MDΔ = 0.062 kg/m^2^; 95% highest posterior density interval (HPD) = −0.293; 0.469] remained unchanged following hazelnut consumption (29–69 g/day; average 45 g/day) for 28–84 days (average 56 days). This meta-analysis was too small to provide accurate data and other metanalyses found no effect. 

Banel et al. 2009 [112] conducted a meta-analysis of 13 studies (13 publications) [96,97,101,102,103,104,113,114,115,116,117,118,119]. Both BMI [mean difference (MD) = −0.4 kg/m^2^; *p* = 0.5] and body weight (MD = −0.05 kg; *p* = 0.97) remained unchanged compared with control diets. Publication bias was not observed [112]. 

Flores-Mateo et al. 2013 [120] undertook a meta-analysis of 33 RCTs published prior to December 2012 to investigate the effects of nut consumption on adiposity in comparison with control diets. The findings showed that nut consumption did not significantly affect body weight (MD = −0.47 kg), BMI (MD = −0.40 kg/m^2^) and waist circumference (MD = −1.25 cm) in comparison with control diets. The analysis of body weight [96,97,101,102,103,104,121,122,123,124,125,126,127,128,129,130,131,132,133,134,135,136,137,138,139,140], BMI [95,96,97,107,114,121,125,127,131,136,137,140,141,142,143] and waist circumference [95,107,125,131,132] came from 26, 15 and 5 publications respectively. Publication bias was not detected in analyses [120].

Mejia et al. 2014 [144] undertook a meta-analysis to examine the effect of tree nuts on waist circumference (15 studies; *n* = 1050) from 15 publications including healthy subjects [109], subjects with dyslipidemia [122], metabolic syndrome [99,105,107,132,137,145,146,147,148] and T2DM [95,125,149,150]. Tree nut consumption did not significantly alter waist circumference compared with a control diet.

None of the meta-analyses showed significant changes in body weight, waist circumference and BMI. The concerns that nut consumption may lead to body weight gain caused by the high-energy density with a high-fat content may be allayed. However, a recent meta-analysis [77] showed significant positive effects of nut consumption on body weight, waist circumference and BMI compared with a control diet. A meta-analysis [77] of 56 intervention studies (55 publications) [95,96,99,100,101,102,103,104,105,110,111,121,122,124,125,126,127,128,129,132,133,134,135,136,137,138,139,140,146,147,151,152,153,154,155,156,157,158,159,160,161,162,163,164,165,166,167,168,169,170,171,172,173,174,175] showed a significant reduction in body weight (WMD = −0.22 kg; 95% CI −0.40, −0.04; I^2^ = 75%). Significant reductions in waist circumference (−0.51 cm; 95% CI −0.95, −0.07; I^2^ = 82%) and BMI (−0.16 kg/m^2^; 95% CI −0.31, −0.01; I^2^ = 76%) have been shown in a data from 23 intervention studies (22 publications) [95,99,105,107,125,132,137,138,146,147,151,153,154,156,159,161,163,164,169,172,173,176] and 39 intervention studies (38 publications) [95,96,105,107,110,111,114,121,125,127,136,137,138,140,141,142,143,146,151,152,153,154,155,156,158,162,163,164,165,166,169,171,172,173,174,175,176,177], respectively. In this meta-analysis, eleven interventions [99,125,139,156,161,166,167,168,169,170,171] were weight interventions with nuts. Several interventions [111,128,132,135,174] and a prospective study [153] and non-randomized study [121] reported weight reduction during an intervention without a weight loss program. It is also noted that this meta-analysis included two prospective studies [125,153]. 

The effects of nut consumption on glycemic control, lipid profiles and blood pressure from meta-analyses of intervention studies are summarized in Table 4. The summary of the effects of nut consumption on inflammation in meta-analyses of intervention studies is shown in Table 5. The effect of nut consumption on endothelial function from meta-analyses of intervention studies is summarized in Table 6. Forest plots for Table 4, Table 5 and Table 6 are presented in Appendix A.

#### 3.2.2. Glycemic Control

Three meta-analyses [3,144,178] of intervention studies have found nuts lowered fasting glucose levels with a reduction ranging from 0.08 to 0.15 mmol/L compared with control diets. In three meta-analyses, control diets were a low-fat diet, a high-fat diet, a libitum diet without nuts, a refined carbohydrate snack, a diet without nuts and a usual diet. The type of nuts were almonds, walnuts, cashews, hazelnuts, cashew and pistachios. The study durations ranged from 4 weeks to 12 months.

A meta-analysis of 20 RCTs [100,103,110,122,132,147,151,172,173,174,177,189,190,191,192,193,194,195,196,197] was conducted by Mazidi et al. 2016 [178]. Nut consumption significantly reduced fasting blood glucose to a small degree (MD = −1.45 mg/dL (0.08 mmol/L); 95% CI −2.20, 0.70; *p* < 0.05).

Mejia et al. 2014 [144] conducted a meta-analysis of 49 RCTs with ≥ 3-week durations (68.8% of studies were < 12-week durations) in 2226 subjects with at least one characteristic of the metabolic syndrome A fasting glucose analysis (26 studies; *n* = 1360) was conducted with otherwise healthy subjects [133,196], subjects with dyslipidemia [198,199], metabolic syndrome [99,105,107,132,137,145,146,147,148,171,200] and T2DM [95,100,125,143,149,150,175,201,202,203]. Tree nut consumption at a median dose of ~50 g/day for a median of 8 weeks significantly reduced fasting blood glucose (Mean difference (MD) = −0.08 mmol/L (95% CI −0.16, −0.01 mmol/L)) compared with control diets. Although statistically significant this is a trivial change clinically.

Viguiliouk et al. 2014 [3] focused on subjects with T2DM in a meta-analysis of 12 RCTs (*n* = 450) of ≥3 weeks duration to investigate the effects of tree nut consumption on glycemic control (glycosylated hemoglobin (HbA1c), fasting glucose, fasting insulin and homeostasis model assessment of insulin resistance index (HOMA-IR)) compared with isocaloric diets without tree nuts. This meta-analysis showed that tree nut consumption at a median dose of 56 g/day over a median follow-up period of ~8 weeks significantly decreased HbA1c (MD = −0.07%; *p* = 0.0003) and fasting glucose (MD = −0.15 mmol/L; *p* = 0.03) compared with control diets. However, fasting insulin and HOMA-IR were not altered. Most RCTs analyzed in this meta-analysis had short study periods and poor study quality [3].

#### 3.2.3. Blood Lipids

In a recent meta-analysis of RCTs by Guasch-Ferre et al. 2018 [94], a diet rich in walnuts significantly reduced total cholesterol (TC) (WMD = −6.99 mg/dL; 95% CI −9.39, −4.58 mg/dL; *p* < 0.001; I^2^ = 0.0%; *p*-het = 0.64), low density lipoprotein cholesterol (LDL-C) (WMD = −5.51 mg/dL; 95% CI −7.72, −3.29 mg/dL; *p* < 0.001; I^2^ = 0.0%; *p*-het = 0.49) and TG (WMD = −4.69; 95% CI −8.93, −0.45; *p* = 0.03; I^2^ = 0.0%; *p*-het = 0.99) by 3.25%, 3.73% and 5.52% respectively compared with control diets. HDL-C was not altered by a diet rich in walnuts compared with control diets (WMD = 0.10 mg/dL; 95% CI −0.78, 0.97 mg/dL; *p* = 0.83; I^2^ = 0.0%; *p*-het = 0.85). Apolipoprotein B (ApoB) was reduced (WMD = −3.74 mg/dL; 95% CI −6.51, −0.97; *p* = 0.008; I^2^ = 0.0%; *p*-het = 0.793) with no significant change in ApoA1 (WMD = −2.91 mg/dL; 95% CI −5.98, 0.08; *p* = 0.057; I^2^ = 0.0%; *p*-het = 0.822) compared with control diets [94].

In this meta-analysis, 24 studies from 23 publications [95,96,97,98,99,100,101,103,104,105,115,116,117,118,119,196,204,205,206,207,208,209,210] including 1020 subjects for TC and LDL-C were analyzed. Twenty-five studies from 24 publications [95,96,97,98,99,100,101,102,103,104,105,106,107,115,116,117,118,119,196,204,206,207,208,210] including 1059 subjects for HDL-C were analyzed. Twenty-four studies from 23 publications [95,96,97,98,99,100,102,103,104,105,107,115,116,117,118,119,196,204,206,207,208,209,210] including 1059 subjects for TG were analyzed. Nine studies from 8 publications (370 subjects) [97,98,99,103,104,115,119,205] for ApoA1 and 11 studies from 10 publications (604 subjects) [97,98,99,103,104,115,119,196,204,205] for ApoB were analyzed.

Similar results were found by Banel et al. 2009 [112] who conducted a meta-analysis of 13 RCTs [96,97,101,102,103,104,113,114,115,116,117,118,119] with a focus on the effects of walnuts (10–24% of total calories; for 4–24 weeks (mean = ~6 weeks) on lipid profiles in comparison with control diets. Walnut consumption significantly reduced TC (MD = −10.3 mg/dL; *p* < 0.001) and LDL-C (MD = −9.2 mg/dL; *p* < 0.001) compared with control diets in 11 studies (11 publications) [96,97,101,103,104,113,115,116,117,118,119]. HDL-C (MD = −0.2 mg/dL; *p* = 0.8) and triglyceride (TG) (MD = −3.9 mg/dL; *p* = 0.3) were not altered compared with control diets.

A meta-analysis of 27 studies (17 parallel and 10 crossover studies) from 18 publications [98,122,125,133,134,137,139,143,146,156,159,162,202,203,211,212,213,214], which was conducted by Musa-Veloso et al. 2016 [179] showed a significantly decrease in fasting, TC, LDL-C and TG but no alteration in HDL-C; −0.017 mmol/L; *p* = 0.207), following almond consumption (20–113 g/day) in periods ranging from 4 weeks to 18 months. With regard to TC levels, 73% of all 27 studies were < 12 weeks long and 63% used ≥ 45 g of almond/day. TC levels from all 27 strata was decreased by −0.153 mmol/L (95% CI −0.235, −0.070 mmol/L; *p* < 0.001). LDL-C in 25 studies from 17 publications [98,122,125,133,134,137,139,143,146,156,159,162,202,203,211,212,214] was reduced by −0.124 mmol/L (95% CI −0.196, −0.051 mmol/L; *p* = 0.001). TG levels from 25 studies from 17 publications [98,122,125,133,137,139,143,146,156,159,162,202,203,211,212,213,214] were decreased by −0.067 mmol/L (95% CI −0.132, −0.002 mmol/L; *p* = 0.042).

A meta-analysis of 20 RCTs [100,103,110,122,132,147,151,172,173,174,177,189,190,191,192,193,194,195,196,197] conducted by Mazidi et al. 2016 [178] showed that nut consumption significantly reduced TC (MD = −0.82 mg/dL), LDL-C (MD = −0.69 mg/dL), HDL-C (MD = 0.54 mg/dL) and ApoAI (MD = 1.38 mg/dL).

Perna et al. 2016 [108] investigated the effects of hazelnut consumption (29–69 g/day; average 45 g/day) for 28–84 days (average 56 days) on lipid profiles using a Bayesian random effect meta-analysis. Hazelnuts in 3 studies [109,110,111] significantly decreased LDL-C (pooled MDΔ = −0.150 mmol/L; 95% highest posterior density interval (−0.308; −0.003).

Gobbo et al. 2015 [180] conducted a meta-analysis of 61 intervention trials (RCTs and non RCTs; *n* = 2582) ranging from 3 to 26 weeks (median 4 weeks) to investigate the effects of tree nuts (walnuts (*n* = 21), almonds (*n* = 16), pistachios (*n* = 7), hazelnuts (*n* = 6), macadamia nuts (*n* = 4), pecans (*n* = 2), cashews (*n* = 2), mixed tree nuts (*n* = 2) and Brazil nuts (*n* = 1)] on TC, LDL-C, HDL-C and TG, Apo A1, ApoB and apo lipoprotein B100 in adults without CVD. An inverse-variance fixed-effects meta-analysis was used to calculate mean differences between nut intervention and control arms, dose-standardized to 1 serving (28.4 g)/d. Nut consumption (per serving/d) significantly decreased TC (−4.7 mg/dL), LDL-C (−4.8 mg/dL), apolipoprotein B (ApoB; −3.7 mg/dL) and TG (−2.2 mg/dL). Tree nut consumption at ≥ 60 g/day showed a stronger effect on TC and LDL-C in a nonlinear manner (*p*-nonlinearity < 0.001) and 100 g tree nuts/day decreased LDL-C by up to 35 mg/dL. The ApoB reduction was greater in subjects with T2DM (−11.5 mg/dL; 95% CI −16.2, −6.8 mg/dL) than in nondiabetic subjects (−2.5 mg/dL; 95% CI −4.7, −0.3 mg/dL) (*p*-het = 0.015).

Mejia et al. 2014 [144] conducted an analysis (44 studies; *n* = 1690) with 42 publications including healthy subjects [110,115,118,133,141,142,196,208,209,212,215,216], subjects with dyslipidemia [104,119,122,127,129,134,214,217], metabolic syndrome [102,105,110,132,137,139,145,146,147,148,171] and T2DM [95,96,100,125,143,149,150,175,201,202,203]. They found a significantly decrease in TG (MD = −0.06 mmol/L (95% CI −0.09, −0.03 mmol/L)) compared with a control diet. HDL-C was not altered after tree nuts consumption compared with a control diet [144].

In this review, the TC-lowering effect of nuts ranged from 0.021 to 0.28 mmol/L and LDL-C was lowered by 0.017 to 0.26 mmol/L comparing nuts to control diets. Interestingly, Banel et al. 2009 [112] found a bigger reduction in TC comparing walnuts with *c*ontrol diets. Part of the reason for the bigger reduction may be related to larger amount of walnuts consumed (30–108 g/day representing 5–25% of total energy).

#### 3.2.4. Adipokines and Inflammatory Markers

A meta-analysis of 32 RCTs (36 publications) [91,95,98,102,103,105,106,110,122,132,147,151,154,167,172,173,174,176,177,190,196,197,205,211,212,218,219,220,221,222,223,224,225,226,227,228]) with ≥ 3-week follow-up period conducted by Neale et al. 2017 [185], showed no significant alteration in adipokines/inflammatory markers including CRP (WMD = −0.01 mg/L; 95% CI −0.06,0.03; *p* = 0.59 26 strata from 25 studies), adiponectin (WMD = 0.29 μg/mL; *p* = 0.53), TNF-α (WMD = −0.05 pg/mL; *p* = 0.17), IL-6 (WMD = −0.02 pg/mL; *p* = 0.65), ICAM-1 (WMD = −0.68 ng/mL; *p* = 0.27) and VCAM-1 (WMD = 2.83 ng/mL; *p* = 0.63).

In a very recent meta-analysis of RCT studies by Xiao et al. 2018 [184], nut consumption was shown to decrease intercellular adhesion molecule 1 (ICAM-1; WMD = −0.17; 95% CI, −0.32, −0.03; *p* = 0.01) from 14 studies (13 publications) [98,103,106,110,151,174,196,205,218,220,223,224,227].

A sub-analysis showed that this decreased effect of nuts was observed in the long-term studies (≥12 weeks) and in parallel RCTs. No significant effects on C-reactive protein (CRP), tumor necrosis factor alpha (TNF-α), vascular cell adhesion molecule (VCAM), IL-6, E-selectin were observed [184].

A meta-analysis of 20 RCTs (20 publications) [100,103,110,122,132,147,151,172,173,174,177,189,190,191,192,193,194,195,196,197] by Mazidi et al. 2016 [178] showed a significant effect of nuts on leptin (WMD = −0.71 mg/dL), with no effect on CRP, IL-6, adiponectin, IL-10 and TNF-α [178]. Biomarkers of endothelial function including E-selectin (WMD = −1.17 ng/L), fibrinogen (WMD = −0.13 pg/mL), ICAM-1 (WMD = −0.12 ng/L) and VCAM-1 (WMD = −0.02 ng/L) were not significantly altered after nut consumption.

Gobbo et al. 2015 [180] also found that tree nut consumption per 1 serving (28.4 g)/day of tree nuts did not significantly alter CRP levels (WMD = 0.1 mg/dL; 95% CI −1.6, 1.8) compared with a control diet [180].

Overall it would appear that nuts have very little effects on inflammatory markers.

#### 3.2.5. Endothelial Function

Large meta-analyses have found a beneficial effect of nut consumption on endothelial function.

Neale et al. 2017 [185] showed a significant improvement in flow-mediated dilation (FMD) (MD = 0.79%; 95% CI 0.35, 1.23; *p* = 0.0004) from 9 studies (8 publications [95,103,105,173,174,176,219,228]; 652 subjects). According to another meta-analysis which was conducted by Xiao et al. 2018 [187] nut consumption significantly improved FMD (MD = 0.41%; 95% CI 0.18, 0.63; *p* = 0.001; I^2^ = 39.5%, *p*-het = 0.094). Walnuts significantly improved FMD (MD = 0.39%; 95% CI 0.16, 0.63; *p* = 0.001).

Huang et al. 2018 [186] examined the effect nut supplementation on FMD showing a significantly increased FMD (MD = 1.03%; 95% CI 0.26, 1.79; *p* = 0.008) in 13 trials.

Fogacci et al. 2017 [188] investigated the effect of pistachios on endothelial reactivity (ER) by conducting a meta-analysis of 4 RCTs (4 publications) [173,174,228,229]. Pistachios did not alter FMD (MD = +0.28%; 95% CI −0.58, 1.13; *p* = 0.525).

#### 3.2.6. Blood Pressure

Meta-analyses of intervention studies have shown no effect of nut consumption on blood pressure.

A recent meta-analysis of RCTs by Guasch-Ferre et al. 2018 [94] showed that a diet rich in walnuts did not alter systolic blood pressure (SBP) (WMD = −0.72 mm Hg; 95% CI −2.75, 1.30; *p* = 0.48) or diastolic blood pressure (DBP) (WMD = −0.10 mm Hg; 95% CI −1.49, 1.30; *p* = 0.88) [94]. Eight studies from 8 publications (402 subjects) [95,98,99,102,103,105,107,206] were analyzed in this meta-analysis [94].

In a meta-analysis of 21 RCTs (1652 subjects; 21 publications) [95,99,101,102,103,115,122,125,127,132,137,139,143,149,191,193,228,229,230,231,232] by Mohammadifard et al. 2015 [181], nut consumption did not significantly alter SBP (MD: −0.91; 95% CI −2.18, 0.36; P = 0.16) overall but lowered it in subjects without T2DM (MD: −1.29 mmHg; 95% CI −2.35, −0.22; *p* = 0.02). In sub-analysis stratified by nut types, only pistachios showed a significant reduction in SBP (MD = −1.82 mmHg; 95% CI −2.97, −0.67; *p* = 0.002), while DBP was significantly reduced by mixed nuts (MD = −1.19 mmHg; 95% CI −2.35, −0.03; *p* = 0.04) and pistachios (MD = −0.80 mmHg; 95% CI −1.43, −0.17; *p* = 0.01) [181]. Mejia et al. 2014 [144] examined the effect of tree nuts on BP (20 studies; *n* = 1267). BP was not significantly changed following tree nut consumption compared with a control diet. Two other meta-analyses [178,180] showed no effect on BP. Gobbo et al. 2015 [180] showed no significant alteration in BP after tree nut consumption (per 1 serving/day) when comparing with a control diet. Mazidi et al. 2016 [178] showed no significant changes in SBP (MD = −0.69 mmHg; 95% CI −1.34, −0.03) and DBP (MD = −0.14 mm Hg; 95% CI −0.54, 0.25).

## 4. Discussion

Nuts are comprised of unique macro and micronutrients (e.g., monounsaturated fatty acids, polyunsaturated fatty acids, fiber, arginine and magnesium) and bioactive compounds (e.g., phytosterols and polyphenols). The synergistic beneficial effects of nuts on cardiometabolic disease result from these nutrient- and non-nutrient compositions of nuts. The potential mechanisms underlying these effects can be explained through the improvement of glucose, lipids, weight, endothelial function and gut microbiota modification [7,233,234]. The most important and most convincingly demonstrated is the reduction in LDL cholesterol which appears to be similar with all nuts and to be clinically significant.

This review highlights findings from meta-analyses of prospective studies that indicate that nut consumption appears to be associated with reduced all-cause mortality, CVD mortality and incidence, coronary heart disease incidence and mortality stroke incidence and mortality. No association between nut consumption and the incidence of T2DM was found in meta-analyses of prospective studies, even though the fasting glucose levels were reduced by a small degree in intervention studies. Moreover, the association between nut consumption and the incidence of hypertension was found in meta-analyses of prospective studies, whereas blood pressure was not favorably altered in meta-analyses of intervention studies.

In this review we have highlighted differences between meta-analyses and explained why they may differ and come to a conclusion about the likely effects of nuts. No other review has focused entirely on meta-analyses both of all nuts and individual nuts. In general, the more recent meta-analyses are larger and hopefully contain all the previous primary studies in earlier meta-analyses but sometimes this is not the case. Where the meta-analyses are reasonably large the results should be similar even if a few studies are added or omitted. All of the meta-analyses on total mortality and cause-specific mortality assessed the study quality using the Newcastle-Ottawa scale or a very similar scale and found no effect of study quality on the outcomes of the meta-analyses. There appears to be little differences between different nut types.

The major areas of new work required are intervention studies to examine the effects of nuts on glucose homeostasis and the incidence of T2DM in at risk populations as well as clarifying the effects on weight and blood pressure as here there are disparities between cohort studies and interventions.

## Figures and Tables

**Figure 1 nutrients-10-01935-f001:**
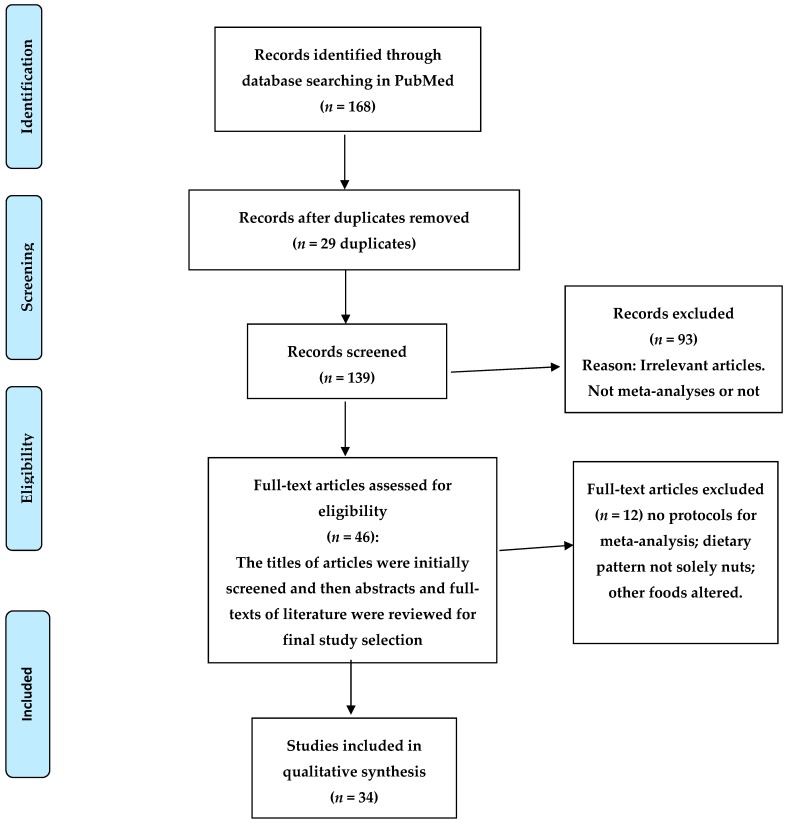
Flow diagram of the literature review.

**Table 1 nutrients-10-01935-t001:** Summary of nut consumption and all-cause, CVD, CHD, stroke, diabetes mortality in meta-analyses of prospective cohort studies.

	Ref/Studies	Analyses	Nut Types	Sample Size	SampleAge (year)	Follow-up Periods (year)	No. of Studies Included	RRs	Significant?
**All-cause mortality**	Chen et al. 2017 [12]/18 prospective studies from 15 publications [17,18,19,20,21,22,23,24,25,26,27,28,29,30,31]	High- vs. Low	total nuts	>498,730 subjects with 66,568 deaths	25–85	4.3–30	16 studies from 13 publications [17,18,19,20,21,22,23,24,25,26,27,28,29]	0.81 (95% CI 0.78, 0.84; *p*-het = 0.20; I^2^ = 22%)	S
			peanuts	265,252 subjects with 44,396 deaths	40–79	1, 10 & 12.2	5 studies from 3 publications [17,18,19]	0.85 (95% CI 0.81, 0.89; *p*-het = 0.19; I^2^ = 34.1%)	S
			tree nuts	130,987 subjects with 36,252 deaths	55–69	4, 10	3 studies from 2 publications [17,19]	0.83 (95% CI 0.77, 0.89; *p*-het = 0.95; I^2^ = 0%)	S
		Dose-response	per 1 serving (28 g) of total nuts per week	>766,470 subjects with 81,034 deaths	25–85	4.3–30	18 studies from 15 publications [17,18,19,20,21,22,23,24,25,26,27,28,29,30,31]	0.96 (95% CI 0.94, 0.97; *p*-het < 0.001; I^2^ = 72.3%)	S
	Schwingshackl et al. 2017 [13]/16 prospective of 14 publications [17,18,19,21,22,23,24,25,26,27,28,29,30,31]	High- vs. Low	total nuts	902,178 subjects with 80,204 deaths	16–87	4.3–30	16 prospective of 14 publications [17,18,19,21,22,23,24,25,26,27,28,29,30,31]	0.80 (95% CI 0.74, 0.86; I^2^ = 84%; *p*-het < 0.001)	S
		Dose-response	per 1 serving (28 g)/day of total nuts	902,178 subjects with 80,204 deaths	16–87	4.3–30	16 prospective of 14 publications [17,18,19,21,22,23,24,25,26,27,28,29,30,31]	0.76 (95% CI 0.69, 0.84; I^2^ = 82%; *p*-het = 0.001)	S
	Aune et al. 2016 [14]/20 prospective studies from 9 publications [17,18,19,20,21,22,23,24,25,26,27,28,29,31,32,33,34,35,36,37,38,39,40,41,42,43,44,45,46]	High- vs. Low	total nuts	819,448 subjects with 85,870 deaths	25–95	2–30	15 studies from 15 publications [17,18,19,20,21,22,23,24,25,26,27,28,29,31,32]	0.81 (95% CI 0.77, 0.85; *p*-het = 0.05; I^2^ = 41%)	S
			peanuts	265,252 subjects with 44,396 deaths	40–79	4–30	5 studies from 3 publications [17,18,19]	0.85 (95% CI 0.82, 0.89; *p*-het = 0.30; I^2^ = 18%)	S
			tree nuts	20,2751 subjects with 42,508 deaths	40–79	4–30	4 studies from 3 publications [17,18,19]	0.80 (95% CI 0.74, 0.86; *p*-het = 0.07; I^2^ = 58%)	S
		Dose-response: 28 g/day increase in total nuts, 10 g/day increase in peanuts and tree nuts	total nuts	819,448 subjects with 85,870 deaths	25–95	2–30	16 studies from 15 publications [17,18,19,20,21,22,23,24,25,26,27,28,29,31,32]	0.78 (95% CI 0.72, 0.84; *p*-het < 0.0001; I^2^ = 66%)	S
			peanuts	265,252 subjects with 44,396 deaths	40–79	4–30	5 studies from 3 publications [17,18,19]	0.77 (95% CI 0.69, 0.86; *p*-het = 0.03; I^2^ = 64%)	S
			tree nuts	202,751 subjects with 42,508 deaths	40–79	4–30	4 studies from 3 publications [17,18,19]	0.82 (95% CI 0.75, 0.90; *p*-het = 0.02; I^2^ = 70%)	S
	Mayhew et al. 2016 [5]/20 Prospective of 20 publications [18,19,20,22,23,24,25,26,33,37,38,39,40,42,43,44,47,48,49,50]	High- vs. Low	total nuts	277,432 subjects with 49,232 deaths	25–94	4.6–30	10 studies from 9 publications [18,19,22,23,24,25,26,49,50]	0.81 (95% CI 0.77, 0.85; *p*-het = 0·04, I^2^ = 43%)	S
		Dose-response	increase 4 servings total nuts/week	277,432 subjects with 49,232 deaths	25–94	4.6–30	10 studies from 9 publications [18,19,22,23,24,25,26,49,50]	0.81 (95% CI 0.75, 0.92)	S
	Grosso et al. 2015 [15]/9 prospective of 9 publications [19,23,24,25,38,39,50,51,52]	High- vs. Low	total nuts	207,608 subjects with 34,482 deaths	≥18	4.8–30	6 studies from 6 publications [19,23,24,25,51,52]	0.77 (95% CI 0.69, 0.87; *p*-het = 0.04; I^2^ = 56%)	S
		Dose-response	1 serving of total nuts/week	263,552 subjects with 30,059 deaths	18–80	4.8–30	5 studies from 5 publications [19,23,25,50,51]	0.96 (95% CI 0.93, 0.98; *p*-het = 0.07; I^2^ = 53%)	S
			1 serving of total nuts/day	263,552 subjects with 30,059 deaths	18–80	4.8–30	5 studies from 5 publications [19,23,25,50,51]	0.73 (95% CI 0.60, 0.88; *p*-het = 0.07; I^2^ = 53%)	S
	Luo et al. 2014 [16]/18 prospective of 18 publications [19,20,22,24,25,26,38,39,40,42,43,44,48,50,53,54,55,56]	High (≥2 to ≥7 servings/week vs. Low (0 to ≤1 serving/week)	total nuts	48,818 deaths	25–94	4–30	7 studies from 7 publications [19,20,22,24,25,26,50]	0.85 (95% CI 0.79, 0.91; *p*-het = 0.005; I^2^ = 60.2%)	S
		Dose-response	total nuts	3,112,510 subjects with 33,595 deaths	40–80	4–30	4 studies from 4 publications [19,22,25,26]	0.83 (95% CI 0.76, 0.91; *p*-het = 0.032; I^2^ = 62.1%)	S
**CVD mortality**	Chen et al. 2017 [12]/18 prospective studies from 15 publications [17,18,19,20,21,22,23,24,25,26,27,28,29,30,31]	High- vs. Low	total nuts	>524,610 subjects with 19,574 deaths	25–85	4.3–30	16 studies from 13 publications [17,18,19,20,21,22,23,25,26,27,28,29,33]	0.75 (95% CI 0.71, 0.79; *p*-het = 0.50; I^2^ = 0%)	S
			peanuts	265,252 subjects with 12,052 deaths	40–79	1, 10&12.2	5 studies from 3 publications [17,18,19]	0.78 (95% CI 0.73–0.85; *p*-het = 0.30; I^2^ = 18%)	S
			tree nuts	130,987 subjects with 9456 deaths	55–69	4, 10	3 studies from 2 publications [17,19]	0.81 (95% CI 0.74, 0.89; *p*-het = 0.62; I^2^ = 0%)	S
		Dose-response	per 1 serving of total nuts per week	>509,871 subjects with 20,362 deaths	25–85	4.3–30	16 studies from 13 publications [17,18,19,20,21,22,25,26,27,28,29,31,33]	0.94 (95% CI 0.93, 0.96; *p*-het = 0.001; I^2^ = 59.9%)	S
	Mayhew et al. 2016 [5]/20 Prospective of 20 publications [18,19,20,22,23,24,25,26,33,37,38,39,40,42,43,44,47,48,49,50]	High- vs. Low	total nuts	243,795 subjects with 13,726 deaths	40–80	4–30	5 studies from 5 publications [17,18,19,22,25]	0.73 (95% CI 0.68, 0.78; *p*-het = 0.31, I^2^ = 16%)	S
		Dose-response	total nuts	243,795 subjects with 13,726 deaths	40–80	4–30	5 studies from 5 publications [17,18,19,22,25]	0.78 (95% CI 0.63, 1.00)	NS
	Grosso et al. 2015 [15]/9 prospective of 9 publications [19,23,24,25,38,39,50,51,52]	High- vs. Low	total nuts	354,933 subjects with 7775 deaths	40–80	4–30	7 studies from 6 publications [19,25,38,39,51,52]	0.71 (95% CI 0.62, 0.81; *p*-het = 0.24; I^2^ = 25%)	S
		Dose-response	1 serving of total nuts/week	354,933 subjects with 7775 deaths	49–80	4–30	4 studies from 4 publications [19,25,38,51]	0.93 (95% CI 0.88, 0.99; *p*-het = 0.004; I^2^ = 74%)	S
			1 serving of total nuts/day	354,933 subjects with 7775 deaths	49–80	4–30	4 studies from 4 publications [19,25,38,51]	0.61 (95% CI 0.42, 0.91; *p*-het = 0.003; I^2^ = 75%)	S
**CHD mortality**	Chen et al. 2017 [12]/18 prospective studies from 15 publications [17,18,19,20,21,22,23,24,25,26,27,28,29,30,31]	High- vs. Low	total nuts	>429,833 subjects with 10,083 deaths	40–80	2–30	13 studies from 10 publications [17,18,19,20,21,22,25,26,27,28]	0.73 (95% CI 0.67, 0.80; *p*-het = 0.30; I^2^ = 14.2%)	S
			peanuts	265,252 subjects with 7025 deaths	40–80	2–30	5 studies from 3 publications [17,18,19]	0.76 (95% CI 0.69, 0.82; *p*-het = 0.65; I^2^ = 0%)	S
			tree nuts	130,987 subjects with 6394 deaths	40–80	2–30	3 studies from 2 publications [17,19]	0.79 (95% CI 0.68, 0.92; *p*-het = 0.25; I^2^ = 27.5%)	S
		Dose-response	per 1 serving of total nuts per week	>412,892 subjects with 10,399 events	40–80	2–30	13 studies from10 publications [17,18,19,20,21,22,25,26,28,31]	0.94 (95% CI 0.93, 0.96; *p*-het = 0.11; I^2^ = 34.3%)	S
	Mayhew et al. 2016 [5]/20 Prospective of 20 publications [18,19,20,22,23,24,25,26,33,37,38,39,40,42,43,44,47,48,49,50]	High- vs. Low	total nuts	278,584 subjects with 8454 events	40–80	2–30	7 studies from 7 publications [17,18,19,22,24,26,39]	0.70 (95% CI 0.64, 0.76; *p*-het = 0.65; I^2^ = 0%)	S
		Dose-response	total nuts	278,584 subjects with 8454 events	40–80	2–30	7 studies from 7 publications [17,18,19,22,24,26,39]	0.78 (95% CI 0.57, 1.08)	NS
**Stroke mortality**	Chen et al. 2017 [12]/18 prospective studies from 15 publications [17,18,19,20,21,22,23,24,25,26,27,28,29,30,31]	High- vs. Low	total nuts	449,293 subjects with 4398 deaths	40–80	2–30	12 studies from 9 publications [17,18,19,21,25,27,28,33,34]	0.82 (95% CI 0.73, 0.91; *p*-het = 0.82; I^2^ = 0%)	S
			peanuts	265,252 subjects with 3315 deaths	40–80	2–30	5 studies from 3 publications [17,18,19]	0.83 (95% CI 0.71, 0.97; *p*-het = 0.12; I^2^ = 46.2%)	S
			tree nuts	130,987 subjects with 2130 deaths	40–80	2–30	3 studies from 2 publications [17,19]	0.93 (95% CI 0.77, 1.13; *p*-het = 0.44; I^2^ = 0%)	NS
		Dose-response	per 1 serving of total nuts per week	>432,352 subjects with 4831 deaths	40–80	2–30	12 studies from 9 publications [17,18,19,21,25,28,31,33,34]	0.95 (95% CI 0.91, 0.997; *p*-het = 0.005; I^2^ = 60.6%)	NS
	Mayhew et al. 2016 [5]/20 Prospective of 20 publications [18,19,20,22,23,24,25,26,33,37,38,39,40,42,43,44,47,48,49,50]	High- vs. Low	total nuts	159,322 subjects with 2166 events	29.2–69	8.3–30	3 studies from 3 publications [17,19,33]	0.83 (95% CI 0.69, 1.00; *p*-het = 0.54; I^2^ = 0%)	NS
		Dose-response	total nuts	159,322 subjects with 2166 events	29.2–69	8.3–30	3 studies from 3 publications [17,19,33]	0.85 (95% CI 0.55, 1.31)	NS
**Diabetes Mortality**	Aune et al. 2016 [14]/20 prospective studies from 9 publications [17,18,19,20,21,22,23,24,25,26,27,28,29,31,32,33,34,35,36,37,38,39,40,41,42,43,44,45,46]	High- vs. Low	total nuts	202,751 subjects with 800 deaths.	40–79	4–30	4 studies from 3 publications [17,18,19]	0.68 (95% CI 0.52–0.90; *p*-het = 0.59; I^2^ = 0%)	S
			peanuts	265,252 subjects with 901 deaths	40–79	4–30	5 studies from 3 publications [17,18,19]	0.84 (95% CI 0.60, 1.19; *p*-het = 0.14; I^2^ = 42.6%)	NS
			tree nuts	130,987 subjects with 4622 deaths	40–79	4–30	3 studies from 2 publications [18,19]	1.19 (95% CI 0.74, 1.89; *p*-het = 0.43; I^2^ = 0%)	NS
		Dose-response	total nuts	202,751 subjects with 800 deaths.	40–79	4–30	4 studies from 3 publications [17,18,19]	0.61 (95% CI 0.43, 0.88; *p*-het = 0.76; I^2^ = 0%)	S
			peanuts	265,252 subjects with 901 deaths	40–79	4–30	5 studies from 3 publications [17,18,19]	0.73 (95% CI 0.45–1.20; *p*-het = 0.32; I^2^ = 15.4%)	NS
			tree nuts	130,987 subjects with 4622 deaths	55–61	10–30	3 studies from 2 publications [17,19]	1.23 (95% CI 0.68, 2.25; *p*-het = 0.62; I^2^ = 0%)	NS

CHD, coronary heart disease; CI, confidence interval; CVD, cardiovascular disease; het, heterogeneity; RCT, randomized-controlled trial; Ref, references; RR, relative risk.

**Table 2 nutrients-10-01935-t002:** Summary of nut consumption and incidence of T2DM, CVD, hypertension, CHD and stroke.

Variables	Ref/Studies	Analyses	Nut Types	Sample Size	Sample Age	Follow-up Periods	No. of Studies Included	RRs	Effects
**T2DM**	Schwingshackl et al. 2017 [60]/8 prospective studies from 7 publications [36,53,54,55,56,61,62]	High vs. Low	total nuts	27,016 events	20–80	4–19.2	8 studies from 7 publications [36,53,54,55,56,61,62]	0.95 (95% CI 0.85, 1.05; *p*-het = 0.004; I^2^ = 67%)	NS
		Dose response: 28 g/day increase	total nuts	27,016 events	20–80	4–19.2	7 studies from 6 publications [36,53,54,55,56,61]	0.89 (95% CI 0.71, 1.12; *p*-het = 0.0002; I^2^ = 77%)	NS
	Wu et al. 2015 [59]/(5 prospective studies from 5 publications [53,54,55,56,84])	High vs. Low	total nuts	263,406 subjects with 11,610 events	20–87	4–19.2	5 studies from 5 publications [53,54,55,56,84]	0.98 (95% CI 0.84, 1.14; *p*-het = 0.004; I^2^ = 74.2%)	NS
	Guo et al. 2015 [63]/8 prospective studies from 8 publications [53,54,55,56,65,68,70,72]	High vs. Low	total nuts	263,663 subjects with 11,580 events	35–77	4.6–23	5 studies from 5 publications [53,54,55,56,65]	0. 98 (95% CI 0.84, 1.15; *p*-het = 0. 008, I^2^ = 67.7%)	NS
		Dose-response:	<1 serving nut per week	263,663 subjects with 11,580 events	35–77	4.6–23	3 studies from 3 publications [53,54,55]	1.00 (95% CI 0.95, 1.04; *p*-het = 0.789, I^2^ = 0.0%)	NS
			1 to 4 servings nut per week	263,663 subjects with 11,580 events	35–77	4.6–23	3 studies from 3 publications [53,54,55]	1.03 (95% CI 0.98, 1.08; *p*-het = 0.903, I^2^ = 0.0%)	NS
			≥5 servings nut per week	263,663 subjects with 11,580 events	35–77	4.6–23	3 studies from 3 publications [53,54,55]	1.04 (95% CI 0.95, 1.14; *p*-het = 0.067, I^2^ = 58.1%)	NS
	Afshin et al. 2014 [57]/16 studies (14 prospective [2,19,22,38,39,40,42,43,47,53,55,56,65,85,86] & 2 RCTs [2,66]	Dose-response: for 4 servings (28.4 g)/week of total nuts	total nuts	230,216 subjects with 13,308 events	35–77	4–19.2	6 studies from 5 publications [53,55,56,65,66]	0.87 (95% CI 0.81, 0.94; *p*-het = 0.269; I^2^ = 21.8%)	S
	Zhou et al. 2014 [64]/23 prospective from 19 publications [22,34,38,40,42,43,48,53,54,55,56,68,70,71,72,86,87,88,89]	High vs. Low	total nuts	342,213 subjects with 14,400 events	34–87	4.6–19.2	6 studies from 5 publications [53,54,55,56,86]	0.92 (95% CI 0.78, 1.09; *p*-het = 0.001; I^2^ = 78.7%)	NS
		Dose-response: 1 serving/day increase	total nuts	342,213 subjects with 14,400 events	34–87	4.6–19.2	6 studies from 5 publications [53,54,55,56,86]	0.80 (95% CI 0.57, 1.14; *p*-het < 0.001; I^2^ = 87.1%)	NS
	Luo et al. 2014 [16]/18 prospective studies from 18 publications [19,20,22,24,25,26,38,39,40,42,43,44,48,50,53,54,55,56]	High (≥2 to ≥7 servings/week) vs. Low (0 to ≤1 serving/week)	total nuts	2,982,852 subjects with 12,655 events	35–72	4	1 study from 1 publication [55]	0.83 (95% CI 0.74, 0.93; *p*-het = 0.278; I^2^ = 14.9%) in multivariable-adjusted model without BMI	S
			total nuts	2,982,852 subjects with 14,486 events	35–77	4–19.2	4 studies from 4 publications [53,54,55,56]	1.00 (95% CI 0.84, 1.19; *p*-het = 0.008; I^2^ = 67.7%) in multivariable-adjusted model with BMI	NS
		Dose-response: 1 serving (28 g)/day increase in total nuts	total nuts	2,982,852 subjects with 14,486 events	35–77	4–19.2	4 studies from 4 publications [53,54,55,56]	0.88 (95% CI 0.84, 0.92; *p*-het = 0.60; I^2^ = 0.0%) after adjustment for age	S
			total nuts	116,4248 subjects with 5121 events	35–77	4	1 study from 1 publication [55]	0.80 (95% CI 0.69–0.94; *p*-het = 0.151; I^2^ = 51.4%) in multivariable-adjusted model without BMI	S
			total nuts	2,918,625 subjects with 13,878 events	35–77	4–19.2	3 studies from 3 publications [53,54,55]	1.03 (95% CI 0.91, 1.16; *p*-het = 0.04; I^2^ = 63.9%) in multivariable-adjusted model with BMI	NS
**Total CVD**	Aune et al. 2016 [14]/20 prospective studies from 29 publications [17,18,19,20,21,22,23,24,25,26,27,28,29,31,32,33,34,35,36,37,38,39,40,41,42,43,44,45,46]	High vs. Low	total nuts	376,228 subjects with 18,655 events	35–79	4–30	12 studies from 12 publications [17,18,19,21,22,25,27,28,29,31,35,36]	0.81 (95% CI 0.74, 0.89; *p*-het = 0.02; I^2^ = 52.3%)	S
			peanuts	265,252 subjects with 12,043 events	40–79	4–30	5 studies from 3 publications [17,18,19]	0.76 (95% CI 0.70, 0.81; *p*-het = 0.90; I^2^ = 0%)	S
			tree nuts	130,987 subjects with 9456 events	55–61	10–30	3 studies from 2 publications [17,19]	0.76 (95% CI 0.69–0.84; *p*-het = 0.92; I^2^ = 0%)	S
		Dose-response: 28 g/day increase in total nuts, 10 g/day increase in peanuts and tree nuts	total nuts	376,228 subjects with 18,655 events	35–89	4–30	12 studies from 12 publications [17,18,19,21,22,25,27,28,29,31,35,36]	0.79 (95% CI 0.70, 0.88; *p*-het = 0.004; I^2^ = 59.6%)	S
			peanuts	265,252 subjects with 12,043 events	40–79	4–30	5 studies from 2 publications [17,18,19]	0.64 (95% CI 0.50–0.81; *p*-het = 0.001; I^2^ = 77%)	S
			tree nuts	130,987 subjects with 9456 events	40–79	4–30	3 studies from 2 publications [17,19]	0.75 (95% CI 0.67–0.84; *p*-het = 0.84; I^2^ = 0%)	S
	Mayhew et al. 2016 [5]/20 prospective studies from 20 publications [18,19,20,22,23,24,25,26,33,37,38,39,40,42,43,44,47,48,49,50]	High vs. Low	total nuts	6309 women with diabetes 634 events	44.3–67.6	22	1 study from 1 publication [48]	0.56 (95% CI 0.36–0.88; *p*-het = 0.54; I^2^ = 0%)	S
		Dose-response: 4 servings/week increase	total nuts	6309 women with diabetes 634 events	44.3–67.6	22	1 study from 1 publication [48]	0.72 (95% CI 0.55–0.96)	S
	Luo et al. 2014 [16]/18 prospective studies from 18 publications [19,20,22,24,25,26,38,39,40,42,43,44,48,50,53,54,55,56]	High (≥2 to ≥7 servings/week) vs. Low (0 to ≤1 serving/week)	total nuts	8862 events	52–80	4.8–30	4 studies from 4 publications [19,22,25,48]	0.70 (95% CI 0.60, 0.81; *p*-het = 0.274; I^2^ = 22.8%)	S
		Dose-response: 1 serving (28 g)/day increase	total nuts	8862 events	52–80	4.8–30	4 studies from 4 publications [19,22,25,48])	0.71 (95% CI 0.59, 0.85; *p*-het = 0.119; I^2^ = 48.8%)	S
**Hypertension**	Schwingshackl et al. 2017 [67]/4 prospective studies from 4 publications [68,69,70,71]	High vs. Low	total nuts	11,962 events	20–95	3–9	4 studies from 4 publications [43,69,70,71]	0.85 (95% CI 0.78, 0.92; I^2^ = 0%; *p*-het = 0.92)	S
		Dose response: 28 g/day increase	total nuts	11,962 events	20–95	3–9	4 studies from 4 publications [43,69,70,71]	0.70 (95% CI 0.45, 1.08; I^2^ = 69%; *p*-het = 0.02)	NS
	Guo et al. 2015 [63]/8 prospective studies from 8 publications [53,54,55,56,65,68,70,72]	High vs. Low	total nuts	30,189 subjects with 9554	34.6–87.1	4.3–15	3 studies from 3 publications [68,70,72]	0.84 (95% CI: 0.76, 0.93; *p*-het = 0.831, I^2^ = 0.0%)	S
		Dose-response:	≥1 serving total nuts per week	30,189 subjects with 9554	34.6–87.1	4.3–15	3 studies from 3 publications [68,70,72]	0. 97 (95% CI 0.83, 1.13; *p*-het =69.2, I^2^= 0.039%)	NS
			≥2 servings total nuts per week	30,189 subjects with 9554	34.6–87.1	4.3–15	3studies from 3 publications [68,70,72]	0. 92 (95% CI 0.87, 0.97; *p*-het = 0.590, I^2^ = 0.0%)	S
	Zhou et al. 2014 [64]/23 prospective from 19 publications [22,34,38,40,42,43,48,53,54,55,56,68,70,71,72,86,87,88,89]	High vs. Low	total nuts	40,102 subjects with 1,2814 events	18–84	4.3–15	4 studies from 4 publications [68,70,71,72]	0.85 (95% CI 0.79, 0.92; *p*-het = 0.927; I^2^ = 0.0%)	S
		Dose-response: 1 serving/day increase	total nuts		18–84	4.3–15	4 studies from 4 publications [68,70,71,72]	0.66 (95% CI 0.44, 1.00; *p*-het = 0.006; I^2^ = 75.9%)	NS
**Total CHD**	Bechthold et al. 2017 [73]/4 prospective studies from 4 publications [37,38,39,40]	High vs. Low	total nuts	143,934 subjects with 5,480 events	30–84	6–26	4 studies from 4 publications [37,38,39,40]	0.80 (95% CI 0.62, 1.03; I^2^ = 79%, *p*-het = 0.002)	NS
	Aune et al. 2016 [14]/20 prospective studies from 29 publications [17,18,19,20,21,22,23,24,25,26,27,28,29,31,32,33,34,35,36,37,38,39,40,41,42,43,44,45,46]	High vs. Low	total nuts	315,397 subjects with 12,331 events	40–80	4–30	11 studies from 10 publications [17,18,19,22,26,27,28,37,38,39]	0.76 (95% CI 0.69, 0.84; *p*-het = 0.04; I^2^ = 47.5%)	S
			peanuts	265,252 subjects with 7025 events	40–79	4–30	5 studies from 3 publications [17,18,19]	0.76 (95% CI 0.69, 0.82; *p*-het = 0.65; I^2^ = 0%)	S
			tree nuts	130,987 subjects with 6394 events	55–61	10–30	3 studies from 2 publications [17,19]	0.79 (95% CI 0.68, 0.92; *p*-het = 0.25; I^2^ = 28%)	S
		Dose-response: 28 g/day increase in total nuts, 10 g/day increase in peanuts and tree nuts	total nuts	315,397 subjects with 12,331 events	40–80	2–30	11 studies from 10 publications [17,18,19,22,26,27,28,37,38,39]	0.71 (95% CI 0.63, 0.80; *p*-het = 0.04; I^2^ = 47.4%)	S
			peanuts	265,252 subjects with 7025 events	40–79	4–30	5 studies from 3 publications [17,18,19]	0.69 (95% CI 0.57, 0.84; *p*-het = 0.12; I^2^ = 45.1%)	S
			tree nuts	130,987 subjects with 6394 events	55–61	10–30	3 studies from 2 publications [17,19]	0.73 (95% CI 0.63, 0.85; *p*-het = 0.44; I^2^ = 0%)	S
	Mayhew et al. 2016 [5]/20 prospective studies from 20 publications [18,19,20,22,23,24,25,26,33,37,38,39,40,42,43,44,47,48,49,50]	High vs. Low	total nuts	123,971 subjects with 4757 events	≥24	10–26	3 studies from 3 publications [37,40,90]	0.66 (95% CI 0.48– 0.91; *p*-het = 0.0002; I^2^ = 88%)	S
	Weng et al. 2016 [58]/14 studies from 10 publications (13 prospective&1 RCT) [20,25,38,39,40,47,48,91,92,93]	High vs. Low	total nuts		30–80	4.8–26	14 studies from 10 publications [20,25,38,39,40,47,48,91,92,93]	0.681 (95% CI 0.592–0.783; *p*-het = 0.001; I^2^ = 62.7%)	S
		Dose-response: 1 serving (28 g)/week increase	total nuts		30–80	4.8–26	13 studies from 9 publications [20,25,38,39,40,47,48,92,93]	0.90 (95% CI 0.87–0.94; *p*-het = 0.000; I^2^ = 68.2%)	S
	Zhou et al. 2014 [64]/23 prospective from 19 publications [22,34,38,40,42,43,48,53,54,55,56,68,70,71,72,86,87,88,89]	High vs. Low	total nuts	179,885 subjects with 7236 events	16–86	3.8–26	9 studies from 7 publications [22,38,40,48,87,88,89]	0.83 (95% CI 0.74, 0.93; *p*-het = 0.010; I^2^ = 59.9%) in risk of CAD	S
		Dose-response: 1 serving (28 g)/day increase	total nuts	179,885 subjects with 7236 events	16–86	3.8–26	9 studies from 7 publications [22,38,40,48,87,88,89]	0.81 (95% CI 0.72, 0.91; *p*-het = 0.018; I^2^ = 56.8%) in risk of CAD	S
	Luo et al. 2014 [16]/18 prospective studies from 18 publications [19,20,22,24,25,26,38,39,40,42,43,44,48,50,53,54,55,56]	High vs. Low	total nuts	6623 events	30–84	7–26	6 studies from 6 publications [22,26,38,39,40,48]	0.66 (95% CI 0.55, 0.78; *p*-het = 0.02; I^2^ = 62.5%)	S
		Dose-response: 1 serving (28 g)/day increase	total nuts	6623 events	30–84	7–26	6 studies from 6 publications [22,26,38,39,40,48]	0.72 (95% CI 0.64, 0.81; *p*-het = 0.644; I^2^ = 0.0%)	S
	Ma et al. 2014 [74] in 13 prospective studies from 9 publications [20,25,38,39,40,47,48,92,93]	High vs. Low	total nuts	493,081 subjects with 6127 events	34–84	4.8–26	13 prospective studies from 9 publications [20,25,38,39,40,47,48,92,93]	0.660 (95% CI 0.581, 0.748; I^2^ = 39.6%) in risk of CAD	S
		Dose response	1 serving/week of nut intake	236,008 subjects with 4886 events	34–80	4.8–22	7 studies from 4 publications [25,38,47,48]	in risk of CAD: 0.96 (95% CI 0.89, 1.02)	NS
			for 2 servings/week of nut intake					0.91 (95% CI 0.82, 0.99)	S
			3 servings/week of nut intake					0.85 (95% CI 0.77, 0.95)	S
			4 servings/week of nut intake					0.80 (95% CI 0.72, 0.89)	S
			5 servings/week of nut intake					0.75 (95% CI 0.65, 0.85)	S
			6 servings/week of nut intake					0.70 (95% CI 0.58, 0.83)	S
**Non-fatal** **CHD/fatal CHD**	Mayhew et al. 2016 [5]/20 prospective studies from 20 publications [18,19,20,22,23,24,25,26,33,37,38,39,40,42,43,44,47,48,49,50]	High vs. Low	total nuts	138,678 subjects with 1565 events	34–84	14–17	3 studies from 3 publications [38,39,47]	Non-fatal CHD: 0.71 (95% CI 0.49– 1.03; *p*-het = 0.03; I^2^ = 72%)	NS
		Dose response: 4 servings/week increase	total nuts	138,678 subjects with 1565 events	34–84	14–17	3 studies from 3 publications [38,39,47]	Non-fatal CHD: 0.81 (95% CI 0.72–0.96)	S
	Afshin et al. 2014 [57]/16 studies (14 prospective [2,19,22,38,39,40,42,43,47,53,55,56,65,85,86]&2 RCTs [2,66]	Dose response: 4 servings (28.4 g)/week	total nuts	141,390 subjects with 2101 events	34–84	14–17	4 studies from 4 publications [2,38,39,47]	Non-fatal CHD: 0.78 (95% CI 0.67–0.92; *p*-het = 0.463; I^2^ = 0.0%)	S
				206,114 subjects with 6749 events	40–84	17–30	6 studies from 5 publications [2,19,22,38,39]	Fatal CHD: 0.76 (95% CI 0.69–0.84; *p*-het = 0.227; I^2^ = 27.7%)	S
**Total stroke**	Bechthold et al. 2017 [73]/4 prospective studies from 4 publications [37,38,39,40]	High vs. Low	total nuts	7490 events	30–84	17–26	6 studies from 5 publications [33,41,42,43,44]	0.94 (95% CI 0.85, 1.05, I^2^ = 18%, *p*-het = 0.30)	NS
		Dose response: 28 g/day increase	total nuts	7490 events	30–84	17–26	6 studies from 5 publications [33,41,42,43,44]	0.99 (95% CI 0.84, 1.17, I^2^ = 45%, *p*-het = 0.11)	NS
	Aune et al. 2016 [14]/20 prospective studies from 29 publications [17,18,19,20,21,22,23,24,25,26,27,28,29,31,32,33,34,35,36,37,38,39,40,41,42,43,44,45,46]	High vs. Low	total nuts	396,768 subjects with 9272 events	40–86.7	4–26	10 studies from 9 publications [17,18,27,28,33,34,41,42,43]	0.89 (95% CI 0.82, 0.97; *p*-het = 0.90; I^2^ = 0%)	S
			peanuts	265,252 subjects with 3315 events.	40–79	4–30	5 studies from 3 publications [18,19,50]	0.83 (95% CI 0.69–1.00; *p*-het = 0.12; I^2^ = 45.9%)	NS
			tree nuts	130,987 subjects with 2130 events	55–61	10–30	3 studies from 2 publications [17,19]	0.93 (95% CI 0.77, 1.13; *p*-het = 0.44; I^2^ = 0%)	NS
		Dose response: 28 g/day increase in total nuts, 10 g/day increase in peanuts and tree nuts	total nuts	396,768 subjects with 9272 events	30–86.7	4–26	11 studies from 9 publications [17,18,27,28,33,34,41,42,43]	0.93 (95% CI 0.83, 1.05; *p*-het = 0.31; I^2^ = 13.7%)	NS
			peanuts	265,252 subjects with 3315 events	40–79	4–30	5 studies from 3 publications [17,18,19]	0.63 (95% CI 0.41, 0.95; *p*-het = 0.001; I^2^ = 77.6%)	S
			tree nuts	130,987 subjects with 2130 events	55–69	10–30	3 studies from 2 publications [19,50]	0.89 (95% CI 0.69, 1.14; *p*-het = 0.58; I^2^ = 0%)	NS
	Mayhew et al. 2016 [5]/20 prospective studies from 20 publications [18,19,20,22,23,24,25,26,33,37,38,39,40,42,43,44,47,48,49,50]	High vs. Low	total nuts	157,826 subjects with 4318 events	30–75	4–26	2 studies from 2 publications [33,42]	1.05 (95% CI 0.69–1.61; *p*-het = 0.04; I^2^ = 77%)	NS
	Afshin et al. 2014 [57]/16 studies (14 prospective [2,19,22,38,39,40,42,43,47,53,55,56,65,85,86]&2 RCTs [2,66]	Dose response: for 4 servings (28.4 g)/week	total nuts	155,685 subjects with 5544 events	30–86.7	4.8–26	4 studies from 3 publications [2,42,43]	0.89 (95% CI 0.74, 1.05; *p*-het = 0.012; I^2^ = 72.7%)	NS
	Zhou et al. 2014 [64]/23 prospective from 19 publications [22,34,38,40,42,43,48,53,54,55,56,68,70,71,72,86,87,88,89]	High vs. Low	total nuts	182,730 subjects with 5669 events	30–86.7	9–26	4 studies from 3 publications [34,42,43]	0.87 (95% CI 0.74, 1.03; *p*-het = 0.317; I^2^ = 15.0%)	NS
		Dose response: 28 g/day increase	total nuts	182,730 subjects with 5669 events	30–86.7	9–26	4 studies from 3 publications [34,42,43]	0.90 (95% CI 0.71, 1.14; *p*-het = 0.114; I^2^ = 49.6%)	NS
	Luo et al. 2014 [16]/18 prospective studies from 18 publications [19,20,22,24,25,26,38,39,40,42,43,44,48,50,53,54,55,56]	High (≥2 to ≥7 servings/week) vs. Low (0 to ≤1 serving/week)	total nuts	6487 events	30–86.7	21–26	3 studies from 3 publications [42,43,44]	0.91 (95% CI 0.81, 1.02; *p*-het = 0.285; I^2^ = 20.4%)	NS
	Shi et al. 2014 [75] in 4 prospective studies from 3 publications [34,42,43]	High vs. Low	total nuts	468,887 subjects with 10,493 events	30–75	9–26	3 publications [34,42,43]	0.90 (95% CI 0.81, 0.99; *p*-het = 0.527, I^2^ = 0)	S

CAD, coronary artery disease; CI, confidence interval; CVD, cardiovascular disease; d, day; HD, coronary heart disease; het, heterogeneity; NS, non-significant; RCT, randomized-controlled trial; Ref, references; RR, relative risk; S, significant.

**Table 3 nutrients-10-01935-t003:** Summary of nut consumption and anthropometric parameters in intervention studies.

References Comparison						
Variables	Li et al. 2018 [77]: nuts vs. controls	Guasch-Ferre et al. 2018 [94]: walnuts vs. controls	Perna et al. 2016 [108]: Bayesian meta-analysis of hazelnuts	Mejia et al. 2014 [144]: ~50 g/day of tree nuts over ~8 weeks vs. controls	Flores-Mateo et al. 2013 [120]: nuts vs. controls	Banel et al. 2009 [112]: walnuts vs. controls
BW	WMD = −0.22 kg (95% CI −0.40, −0.04) From 56 RCTs (55 publications) [95,96,99,100,101,102,103,104,105,110,111,121,122,124,125,126,127,128,129,132,133,134,135,136,137,138,139,140,146,147,151,152,153,154,155,156,157,158,159,160,161,162,163,164,165,166,167,168,169,170,171,172,173,174,175]	WMD = −0.12 kg (95% CI −2.12, 1.88; *p* overall effect = 0.90) From 10 RCTs [95,96,97,98,99,100,101,102,103,104]			WMD = –0.47 kg (95% CI −1.17, 0.22; I^2^ = 7%; *p* overall effect = NS) From 26 RCTs [96,97,101,102,103,104,121,122,123,124,125,126,127,128,129,130,131,132,133,134,135,136,137,138,139,140]	WMD = −0.05 kg (*p* overall effect = 0.97) No 95% CI reported. From 13 RCTs [96,97,101,102,103,104,113,114,115,116,117,118,119]
WC	WMD = −0.51 cm (95% CI −0.95, −0.07) From 23 RCTs (22 publications) [95,99,105,107,125,132,137,138,146,147,151,153,154,156,159,161,163,164,169,172,173,176]			MD = −0.62 cm (95% CI −1.54, 0.30; I^2^= 67%; *p*-het < 0.0001; *p* overall effect = 0.19) From 15 RCTs [95,99,105,107,109,122,125,132,137,145,146,147,148,149,150]	WMD = –1.25 cm (95% CI −2.82, 0.31; I^2^ = 28%; *p* overall effect = NS) From 5 RCTs [95,107,125,131,132]	
BMI	WMD = −0.16 kg/m^2^ (95% CI −0.31, −0.01) From 39 RCTs (38 publications) [95,96,105,107,110,111,114,121,125,127,136,137,138,140,141,142,143,146,151,152,153,154,155,156,158,162,163,164,165,166,169,171,172,173,174,175,176,177]	WMD = −0.11 kg/m^2^ (95% CI −1.15, 0.92; *p* overall effect = 0.82) From 6 RCTs [95,96,97,105,107,114]	MDΔ = 0.062 kg/m^2^ (95% HPD −0.293, 0.469) From 3 RTCs [109,110,111]		WMD = −0.40 kg/m^2^ (95% CI −0.97, 0.17; I^2^ = 49%; *p* overall effect = NS) From 15 RCTs [95,96,97,107,114,121,125,127,131,136,137,140,141,142,143]	WMD = −0.4 kg/m^2^ (*p* overall effect = 0.5) No 95% CI reported. From 13 RCTs [96,97,101,102,103,104,113,114,115,116,117,118,119]

BW body weight, BMI, body mass index; het, heterogeneity; 95% HPD, 95% highest posterior density interval; MDΔ, mean differences of Delta-changes from baseline across treatment; NS, non-significant; T2DM, type 2 diabetes mellitus; WC, waist circumference, WMD; weighted mean difference.

**Table 4 nutrients-10-01935-t004:** Summary of nut consumption and glucose, insulin, lipids and blood pressure in intervention studies.

References Comparison											
Variables	Guasch-Ferre et al. 2018 [94]: walnuts vs. controls	Mazidi et al. 2016 [178]: nuts vs. controls	Musa-Veloso et al. 2016 [179]: almonds vs. controls	Perna et al. 2016 [108]: Bayesian meta-analysis of hazelnuts	Gobbo et al. 2015 [180]: per 1 serving/day of tree nuts vs. controls	Mohammadifard et al. 2015 [181]: nuts vs. controls	Mejia et al. 2014 [144]: ~50 g/day of tree nuts over ~8 weeks vs. controls	Viguiliouk et al. 2014 [3]: tree nuts vs. controls in subjects with T2DM	Sabate et al 2010 [182]: 67 g/day of nuts vs. controls	Phung et al. 2009 [183]: almonds (25–169 g/day) vs. controls	Banel et al. 2009 [112]: walnuts vs. controls
FBG		−1.45 mg/dL (95% CI −2.20, 0.70; *p* overall effect < 0.05) =−0.08 mmol/L (95% CI −0.12, 0.039 mmol/L)					−0.08 mmol/L (95% CI −0.16, −0.01; I^2^ = 41%; *p*-het = 0.02; *p* overall effect = 0.03)	−0.15 mmol/L (95% CI −0.27, −0.02; I^2^ = 35%; *p*-het = 0.12; *p* overall effect = 0.03)			
Fasting Insulin								−3.42 pmol/L (95% CI −10.06, 3.21; I^2^ = 72%; *p*-het = 0.0004; *p* overall effect = 0.031)			
HOMA-IR								−0.24 (95% CI −0.51, 0.04; I^2^ = 87%; *p*-het = 0.0005; *p* overall effect = 0.10)			
HbA1c								−0.07% (−95% CI 0.10, −0.03; I^2^ = 37%; *p*-het = 0.13; *p* overall effect = 0.0003)			
TC	−6.99 mg/dL; (95% CI −9.39, −4.58 mg/dL; *p* overall effect < 0.001; I^2^ = 0.0%; *p*-het = 0.64) =−0.18 mmol/L (95% CI −0.24, −0.118 mmol/L)	−0.82 mg/dL (95% CI −1.53 mg/dL, −0.11; *p* overall effect < 0.05) = −0.021 mmol/L (95% CI −0.04, −0.003 mmol/L)	−0.153 mmol/L (95% CI −0.235, −0.070 mmol/L; *p* overall effect < 0.001)	MDΔ = −0.127 mmol/L (95% HPD −0.284, 0.014 mmol/L)	−4.7 mg/dL (95% CI −5.3, −4.0 mg/dL; I^2^ > 30%; *p*-het = 0.001; *p* overall effect < 0.05) = −0.12 mmol/L (95% CI −0.14, −0.1 mmol/L)				−10.9 mg/dL (95% CI −14.1, −7.8 mg/dL) (5.1% change; *p* < 0.001) = −0.28 mmol/L (95% CI −0.36, −0.2 mmol/L)	−0.18 mmol/L (95% CI −0.34, −0.02 mmol/L; I^2^ = 0.0%; *p*-het = NS; *p* overall effect = 0.030)	−10.3 mg/dL (95% CI −14.76, −5.83 mg/dL; I^2^ = 0.0%; *p*-het = 0.63; *p* overall effect < 0.001) = −0.26 mmol/L (95% CI −0.38, −0.15 mmol/L)
LDL-C	−5.51 mg/dL (95% CI −7.72, −3.29 mg/dL; *p* overall effect < 0.001; I^2^ = 0.0%; *p*-het = 0.49) =−0.14 mmol/L (95% CI −0.2, −0.085 mmol/L)	−0.69 mg/dL (95% CI −1.32, −0.07; *p* overall effect < 0.05) = −0.017 mmol/L (95% CI −0.03, −0.002 mmol/L)	−0.124 mmol/L (95% CI −0.196, −0.051 mmol/L; *p* overall effect = 0.001)	MDΔ = −0.150 mmol/L (95% HPD −0.308, −0.003)	−4.8 mg/dL (95% CI −5.5, −4.2 mg/dL; I^2^> 30%; *p*-het = 0.01; *p* overall effect < 0.05) = −0.12 mmol/L (95% CI −0.14 −0.11 mmol/L)				−10.2 mg/dL (95% CI −13.1, −7.4 mg/dL; 7.4% change; *p* overall effect < 0.001) = −0.26 mmol/L (95% CI −0.34, −0.19 mmol/L)	−0.15 mmol/L (95% CI −0.29, 0.0001 mmol/L; I^2^ = 0.0%; *p*-het = NS; *p* overall effect = 0.05)	−9.2 mg/dL (95% CI −13.1, −5.36 mg/dL; I^2^ = 0.0%; *p*-het = 0.65; *p* overall effect < 0.001) = −0.24 mmol/L (95% CI −0.34, −0.138 mmol/L)
HDL-C	0.10 mg/dL (95% CI −0.78, 0.97 mg/dL; *p* overall effect = 0.83; I^2^ = 0.0%; *p*-het = 0.85) = 0.003 mmol/L (95% CI −0.02, 0.025 mmol/L)	0.54 mg/dL (95% CI 0.17, 0.90; *p* overall effect < 0.05) 0.014 mmol/L (95% CI 0.004, 0.023 mmol/L)	−0.017 mmol/L (95% CI −0.043 mmol/L, 0.009; *p* overall effect = 0.207)	MDΔ = 0.002 mmol/L (95% HPD −0.140, 0.147 mmol/L)	−0.3 mg/dL (95% CI −0.9, 0.4; I^2^> 30%; *p*-het = 0.33; *p* overall effect = NS) −0.007 mmol/L (95% CI −0.02, 0.01 mmol/L)		0.00 mmol/L (95% CI −0.01, 0.01 mmol/L; I^2^ = 86%; *p*-het < 0.00001; *p* overall effect = 0.93).		0.09 mg/dL (95% CI −1.00, 1.19 mg/dL; *p* overall effect = NS) = 0.002 mmol/L (95% CI −0.026, 0.03 mmol/L)	−0.05 mmol/L (95% CI −0.10, 0.01 mmol/L; I^2^ = 0.0%; *p*-het = NS; *p* overall effect= 0.08)	−0.2 mg/dL (95% CI −1.79, 1.38 mg/dL; I^2^ = 0.0%; *p*-het = 0.8; *p* overall effect = 0.8) =−0.005 mmol/L (95% CI −0.05, 0.036 mmol/L)
TG	−4.69 mg/dL (95% CI −8.93, −0.45 mg/dL; *p* overall effect = 0.03; I^2^ = 0.0%; *p*-het = 0.99) =−0.052 mmol/L (95% CI −0.1, −0.005 mmol/L)	−0.66 mg/dL (95% CI −1.34, 0.01; *p* overall effect = NS) =−0.007 mmol/L (95% CI −0.015, 0.00 mmol/L)	−0.067 mmol/L (95% CI −0.132, −0.002 mmol/L; *p* overall effect = 0·042)	MDΔ = 0.045 mmol/L (95% HPD −0.195, 0.269)	−2.2 mg/dL (95% CI −3.8, −0.5mg/dL; I^2^> 30%; *p*-het = 0.16; *p* overall effect < 0.05) =0.02 mmol/L (95% CI −0.04, −0.006 mmol/L)		−0.06 mmol/L (95% CI −0.09, −0.03; I^2^ = 34%; *p*-het = 0.02; *p* overall effect < 0.0001)		−20.6 mg/dL (10.2% change; (95% CI −30.7, −9.9 mg/dL; *p* overall effect < 0.05) in subjects with TG of 150 mg/dL. = −0.23 mmol/L (95% CI −0.3, −0.1 mmol/L)	−0.04 mmol/L (95% CI −0.20, 0.11 mmol/L; I^2^ = 0.0%; *p*-het = NS; *p* overall effect = 0.58)	−3.9 mg/dL (95% CI −11.92, 4.20 mg/dL; I^2^ =0.0%; *p*-het = 0.99; *p* overall effect = 0.3), =−0.04 mmol/L (95% CI −0.13, 0.05 mmol/L)
ApoA1	−2.91 mg/dL (95% CI −5.98, 0.08 mg/dL; *p* overall effect = 0.057; I^2^ = 0.0%; *p*-het = 0.822)	1.38 mg/dL (95% CI 0.15, 2.61 mg/dL; *p* overall effect < 0.05)			−0.6 mg/dL (95% CI −1.9, 0.7 mg/dL; I^2^> 30%; *p*-het = 0.38; *p* overall effect = NS)						
TC: HDL-C			−0.207 (95% CI −0·362, −0.052; *p* overall effect = 0.009)						−0.24 (5.6% change; *p* overall effect = 0.001)		
LDL-C: HDL-C			−0.089 (95% CI −0·209, 0.031; *p* overall effect = 0.145)						−0.22 (8.3% change; *p* overall effect = 0.001)	−0.04 (95% CI −0.21, 0.14; I^2^ = 0.0%; *p*-het = NS; *p* overall effect = 0.670)	
ApoB	−3.74 mg/dL; 95% CI −6.51, −0.97 mg/dL; *p* overall effect = 0.008; I^2^ = 0.0%; *p*-het = 0.793)				−3.7 mg/dL (95% CI −5.2, −2.3 mg/dL; I^2^ > 30%; *p*-het = 0.17; *p* overall effect < 0.05)						
ApoB100		−1.50 mg/dL (95% CI −2.43, 0.57; *p* overall effect = NS)			−2.8 mg/dL (95% CI −6.2, 0.7 mg/dL; I^2^> 30%; *p*-het = 0.31; *p* overall effect = NS)						
SBP	−0.72 mmHg (95% CI −2.75, 1.30 mmHg; *p* overall effect = NS)	−0.69 mmHg (95% CI −1.34 mmHg, 0.03; *p* overall effect = NS)			0.3 mmHg (95% CI −0.8, 1.4 mmHg; I^2^ > 30%; *p*-het = 0.001; *p* overall effect = NS)	−0.91 mmHg (95% CI −2.18, 0.36 mmHg; I^2^ = 73.8% *p*-het < 0.001; *p* overall effect = NS) for all type nuts. −1.29 mmHg (−2.35, −0.22; I^2^ = 53.3%; *p*-het = 0.002; *p* overall effect = 0.02) for all type nuts ^#^. −1.82 mmHg (−2.97, −0.67; *p*-het: NS *p* overall effect = 0.002) for pistachios	0.07 mmHg (95% CI −1.54, 1.69 mmHg; I^2^ = 64%; *p*-het < 0.0001; *p* overall effect = NS)				
DBP	0.10 mmHg (95% CI −1.49, 1.30 mmHg; *p* overall effect= NS)	−0.14 mmHg (95% CI −0.54, 0.25 mmHg; *p* overall effect = NS)			0.4 mmHg (95% CI −0.8, 1.6 mmHg; I^2^> 30%; *p*-het = 0.32; *p* overall effect = NS)	0.21 mmHg (95% CI −0.54, 0.97 mmHg; I^2^ = 69.6%; *p*-het < 0.001; *p* overall effect = NS) for all type nuts. −1.19 mmHg (95% CI −2.35, −0.03; *p* overall effect = 0.04; *p*-het = NS) for mixed nuts. −0.80 mmHg (95% CI −1.43, −0.17; *p*-het = 0.44; I^2^ = 0.0%; *p* overall effect = 0.04) for pistachios ^#^	0.23 mmHg (95% CI −0.38, 0.83 mmHg; I^2^ = 34%; *p*-het = 0.07; *p* overall effect = NS)				

^#^: only in subjects without T2DM. ApoA1, apolipoprotein A1; ApoB, apolipoprotein B; ApoB100, apolipoprotein B100; CRP, C-reactive protein; DBP, diastolic blood pressure; FBG, fasting blood glucose; HOMA-IR, homeostasis model assessment of insulin resistance index; het, heterogeneity; HDL-C, high density lipoprotein cholesterol; 95% HPD, 95% highest posterior density interval; ICAM−1, intracellular adhesion molecule-1, IL-6, interleukin-6; IL-10, Interleukin-10; LDL-C, low density lipoprotein cholesterol; MDΔ, mean differences of Delta-changes from baseline across treatment; NS, non-significant; RCT, randomized controlled trial; SBP, systolic blood pressure; TC, total cholesterol; TG, triglyceride; T2DM, type 2 diabetes mellitus; TNF-α, tumor necrosis factor-alpha; VCAM-1, vascular cell adhesion molecule-1.

**Table 5 nutrients-10-01935-t005:** Summary of nut consumption and adipokines and inflammatory markers in intervention studies.

References Comparisons				
Variables	Xiao et al. 2018 [184] nuts vs. controls	Neal et al. 2017 [185] nuts vs. controls	Mazidi et al. 2016 [178]: nuts vs. controls	Gobbo et al. 2015 [180]: per 1 serving/day of tree nuts vs. controls
Leptin			WMD = −0.71 mg/dL (95% CI −1.11, −0.30, I^2^ = 6.3%; *p* overall effect < 0.05)	
Adiponectin		WMD = 0.29 μg/mL (95% CI −0.63, 1.21; I^2^ = 79% *p*-het < 0.0001; *p* overall effect = NS) WMD = 0.029 mg/dL (95% CI −0.063, 0.121 mg/dL)	WMD = −0.60 mg/dL (95% CI −1.88, 0.68 mg/dL, I^2^ = 5.6%; *p* overall effect = NS)	
CRP	WMD = −0.09 mg/L (95% CI −0.21, 0.02 mg/L; I^2^ < 50%; *p*-het = NS; *p* overall effect = NS)	WMD = −0.01 mg/L (95% CI −0.06,0.03 mg/L; I^2^ = 20%; *p*-het = 0.18; *p* overall effect = NS)	WMD = 0.17 mg/L (95% CI −0.67, 0.33 mg/L; I^2^ = 52.1%; *p* overall effect = NS)	WMD = 0.1 mg/dL (95% CI −1.6, 1.8; I^2^ > 30%; *p*-het = 0.84; *p* overall effect = NS) WMD = 1 mg/L (95% CI −16, 18 mg/L)
IL-6	WMD = −0.09 pg/mL (95% CI −0.24, 0.04; I^2^ < 50%; *p*-het = NS; *p* overall effect = NS)	WMD = −0.02 pg/mL (95% CI −0.12, 0.08; I^2^ = 10% *p*-het = 0.34; *p* overall effect = NS)	WMD = −0.06 ng/dL (95% CI −0.69, 0.56, I^2^ = 9.6%; *p* overall effect = NS) WMD = −0.6 pg/mL (95% CI −6.9, 5.6 pg/mL)	
IL-10			WMD = −0.18 mg/dL (95% CI −1.24, 0.88, I^2^ = 9.3%; *p* overall effect = NS)	
TNF-α	WMD = −0.19 ng/L (95% CI −0.41, 0.03; I^2^ < 50%; *p*-het = NS; *p* overall effect = NS) WMD = −0.19 pg/mL (95% CI −0.41, 0.03 pg/mL)	WMD = −0.05 pg/mL (95% CI −0.13, 0.002; I^2^ = 2% *p*-het = 0.42; *p* overall effect = NS)	WMD = −0.37 pg/mL (95% CI −0.90, 0.16, I^2^ = 7.9%; *p* overall effect = NS)	
ICAM-1	WMD = −0.17 ng/mL (95% CI −0.32, −0.03 ng/mL; I^2^ < 50%; *p*-het = NS; *p* overall effect = 0.01)	WMD = 0.68 ng/mL (95% CI −0.53, 1.89 ng/mL; I^2^ = 0% *p*-het = 0.82; *p* overall effect = NS)	WMD = −0.12 ng/L (95% CI −0.43, 0.18 ng/L; *p* overall effect = NS) WMD = −0.00012 ng/mL (95% CI −0.00043, 0.00018 ng/mL; *p* overall effect = NS)	
VCAM-1	WMD = −0.12 ng/mL (95% CI −0.26, 0.02 ng/mL; I^2^ < 50%; *p*-het = NS; *p* overall effect = NS)	WMD = 2.83 ng/mL (95% CI −8.85, 14.51 ng/mL; I^2^ = 0% *p*-het = 0.70; *p* overall effect = NS)	WMD = −0.02 ng/L (95% CI −0.33, 0.29 ng/L; *p* overall effect = NS) WMD = −0.00002 ng/mL (95% CI −0.00033, 0.00029 ng/mL)	
Fibrinogen			WMD = −0.13 pg/mL (95% CI −1.43, 1.70; *p* overall effect = NS)	
E-selectin	WMD = −0.18 ng/mL (95% CI −0.38, 0.01; I^2^ < 50%; *p*-het = NS; *p* overall effect = NS)		WMD = −1.17 ng/L (95% CI −2.40, 0.06; *p* overall effect = NS) WMD = −0.00117 ng/mL (95% CI −0.0024, 0.00006 ng/mL)	

CI, confidence interval; CRP, C-reactive protein; het, heterogeneity; ICAM-1, intercellular adhesion molecule 1; IL-6, interleukin-6; IL-10, Interleukin-10; NS, non-significant; TNF-α, tumor necrosis factor-alpha; VCAM-1, vascular cell adhesion molecule-1, WMD, weighted mean difference.

**Table 6 nutrients-10-01935-t006:** Summary of nut consumption and endothelial function in intervention studies.

References Comparisons				
Variables	Huang et al. 2018 [186]	Xiao et al. 2018 [187]	Fogacci et al. 2017 [188]	Neale et al. 2017 [185] nuts vs. controls
FMD	1.03% (95% CI 0.26, 1.79; *p* overall effect = 0.008)	0.41% (95% CI 0.18, 0.63; *p* overall effect = 0.001; I^2^ = 39.5%, *p*-het = 0.094)	+0.28% (95% CI −0.58, 1.13; *p* overall effect = 0.525)	0.79% (95% CI 0.35, 1.23; I^2^ = 0%; *p*-het = 0.45; *p* overall effect = 0.0004)
BAD			+0.04% (95% CI 0.03, 0.06; *p* overall effect < 0.001)	

BAD, brachial artery diameter; CI, confidence interval; ER, endothelial reactivity; FMD, flow-mediated dilation; het, heterogeneity.

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
