# Peer review of "Nuts and Cardio-Metabolic Disease: A Review of Meta-Analyses"

_nutrients, 2018, doi:10.3390/nu10121935_

Reviewer 1 Report

“Nuts and Cardio-metabolic Disease: A Systematic Review of Meta-Analyses”

This review is of interests; however, there are some concerns that authors need to address to improve the quality and the impact of their conclusions. The study is a simple literature summary.

Introduction: This is written in very limited way and written part is also not impressive. I suggest the authors should start with background of cardiometabolic, nuts, and then diabetes as one of cardiometabolic disease.

Methods: Please revise; start with how the literature initially was selected? Add additional considerations of inclusion criteria. Please revise the PRISMA flow diagram.

Results: Please write 1-2 sentences, what results want to be summarized? Tables are too long, is it possible to summarize and use “Forest Plot”? This will make the review more impactful.

Discussion: Authors are suggested to including a discussion on how the current study is different from others, please highlighted this more precisely.

Author Response

Comments and Suggestions for Authors

“Nuts and Cardio-metabolic Disease: A Systematic Review of Meta-Analyses”

This review is of interests; however, there are some concerns that authors need to address to improve the quality and the impact of their conclusions. The study is a simple literature summary.

Introduction: This is written in very limited way and written part is also not impressive. I suggest the authors should start with background of cardiometabolic, nuts, and then diabetes as one of cardiometabolic disease.

 Thank you. A sentence on cardiometabolic disease has been written at lines 35-37. We have also added a sentence on why this review was done

Methods: Please revise; start with how the literature initially was selected? Add additional considerations of inclusion criteria. Please revise the PRISMA flow diagram.

Thank you. The sentences have been written in the flow diagram.

“The titles of articles were initially screened and then abstracts and full-texts of literature were reviewed for final study selection”.

Thank you. The sentence has been written in the text at lines 60-64

“Inclusion criteria were nuts as outlined above, meta-analysis, all-cause mortality, mortality, incidence, CVD, coronary heart disease (CHD), stroke, T2DM, hypertension, metabolic syndrome, obesity, adiposity, glycemic control, glucose, lipids, blood pressure, inflammatory markers, endothelial function, flow-mediated dilation.”

 Results: Please write 1-2 sentences, what results want to be summarized? Tables are too long, is it possible to summarize and use “Forest Plot”? This will make the review more impactful.

Thank you. One to two sentences written on results in previous discussion have been moved to each result section for summary of each result. Forest Plots have been added with supplementary materials.

Discussion: Authors are suggested to including a discussion on how the current study is different from others, please highlighted this more precisely.

In the introduction we have highlighted where there is confusion in the literature with differing results in different meta-analyses 

Reviewer 2 Report

I wish to thank the reviewers for their hard work on this very interesting topic. However, I do have concerns about some methodologic aspects refraining me to approve the term “systematic” review for this manuscript. Here are some comments to improve the manuscript and systematize your work; in case of non-acceptance of these suggestions, I would be more comfortable with the term “review”. The main reason for this suggestion is due to the lack of clearly defined outcomes before the onset of study selection, which shows a more “explanatory” element of the review not concordant with systematic reviews per se. Further, other elements are lacking such as study selection and data collection in duplicate and limited number of databases searched .

Title

This is a review of systematic reviews and meta-analyses, please adjust the text accordingly

 Abstract:

Accumulating epidemiological and intervention evidence suggests that nut consumption is associated with reduced incidence cardiometabolic diseases

The purpose of doing a systematic review is because a question has not been yet answered, not because no systematic review has been performed yet... Otherwise it does not convince me of the importance of your topic.

This is a review of systematic reviews and meta-analyses, please adjust the text accordingly

Introduction

This sentence is not relevant to the text unless you discuss the mechanisms. I suggest you remove it: A previous review article [10] addressed possible mechanisms of actions of nuts from human and animal studies.

Methods

The methodology presented here is not in concordance with PRISMA other than the flow diagram... Please refer to the PRISMA checklist for systematic reviews. Here are the missing elements in order to call this review a systematic review:

Please provide the registration number of the review, otherwise I suggest to clearly state the absence of protocol registration in the review

Dates of coverage of databases

Language of publications

Publication status (abstract, articles in peer-reviewed journals, etc.)

Eligibility criteria about the intervention (which kind of nuts, administration, etc.) and population must be clearly defined (must be defined a priori)

How was the data collection was performed? (e.g. with a piloted data collection form)

Which data did you collected in systematic reviews included? (must be defined a priori)

Primary and secondary outcomes should be clearly stated in the review... In systematic reviews, outcomes should be clearly stated and defined a priori...Defining outcomes a post priori inevitably create bias in study selection. In case the cardiometabolic outcomes were determined after the study selection, this review should not be a “systematic review” but a “review” solely.

Searching PUBMED solely is not sufficient nowadays for systematic reviews. At least EMBASE and WEB OF SCIENCES must be searched together with PUBMED, otherwise the review is hardly systematic.

Results

There are a lot of results, which makes the reading difficult and hard to associate with the following discussion. I suggest merging the discussion with the result section, starting with systematic reviews of observational studies a of given condition, followed by intervention studies and then by interpretation of results. Further, most of the discussion could easily go in the results section since it is mainly summarizing the results

Discussion

I have some concerns about potential overlapping of studies in included systematic reviews and how it could influence data interpretation. Please discuss.

Please comment the quality of evidence and risk of bias of studies in the included systematic reviews. If quality of evidence is low, it is impossible to conclude. I suggest you analyse the quality of evidence of included reviews and change the conclusion with a more moderated statement.

 Author Response

Comments and Suggestions for Authors

I wish to thank the reviewers for their hard work on this very interesting topic. However, I do have concerns about some methodologic aspects refraining me to approve the term “systematic” review for this manuscript. Here are some comments to improve the manuscript and systematize your work; in case of non-acceptance of these suggestions, I would be more comfortable with the term “review”. The main reason for this suggestion is due to the lack of clearly defined outcomes before the onset of study selection, which shows a more “explanatory” element of the review not concordant with systematic reviews per se. Further, other elements are lacking such as study selection and data collection in duplicate and limited number of databases searched .

Title

This is a review of systematic reviews and meta-analyses, please adjust the text accordingly

 Thank you. The title has been revised.

“Nuts and Cardio-metabolic Disease: A Review of Meta-Analyses”

Abstract:

Accumulating epidemiological and intervention evidence suggests that nut consumption is associated with reduced incidence cardiometabolic diseases

Thank you. The word of incidence has been inserted.

 The purpose of doing a systematic review is because a question has not been yet answered, not because no systematic review has been performed yet... Otherwise it does not convince me of the importance of your topic.

This is a review of systematic reviews and meta-analyses, please adjust the text accordingly

Thank you. The words of systematic review have been deleted through the texts. We have highlighted in the introduction why this review was done

 Introduction This sentence is not relevant to the text unless you discuss the mechanisms. I suggest you remove it: A previous review article [10] addressed possible mechanisms of actions of nuts from human and animal studies.

Thank you. The sentence has been removed.

Methods The methodology presented here is not in concordance with PRISMA other than the flow diagram... Please refer to the PRISMA checklist for systematic reviews. Here are the missing elements in order to call this review a systematic review:

Please provide the registration number of the review, otherwise I suggest to clearly state the absence of protocol registration in the review

Dates of coverage of databases

Language of publications

Publication status (abstract, articles in peer-reviewed journals, etc.)

Eligibility criteria about the intervention (which kind of nuts, administration, etc.) and population must be clearly defined (must be defined a priori)

How was the data collection was performed? (e.g. with a piloted data collection form)

Which data did you collected in systematic reviews included? (must be defined a priori)

Primary and secondary outcomes should be clearly stated in the review... In systematic reviews, outcomes should be clearly stated and defined a priori...Defining outcomes a post priori inevitably create bias in study selection. In case the cardiometabolic outcomes were determined after the study selection, this review should not be a “systematic review” but a “review” solely.

Searching PUBMED solely is not sufficient nowadays for systematic reviews. At least EMBASE and WEB OF SCIENCES must be searched together with PUBMED, otherwise the review is hardly systematic.

For a review of meta-analyses many of the PRISMA criteria for systematic reviews of intervention and cohort studies are not relevant and we have deleted the section about following PRISMA guidelines.

Results There are a lot of results, which makes the reading difficult and hard to associate with the following discussion.

I suggest merging the discussion with the result section, starting with systematic reviews of observational studies a of given condition, followed by intervention studies and then by interpretation of results. Further, most of the discussion could easily go in the results section since it is mainly summarizing the results

Thank you. Most of the discussion has been moved to the results section in a summary form.

 Discussion I have some concerns about potential overlapping of studies in included systematic reviews and how it could influence data interpretation. Please discuss.

Inevitably there will be considerable overlap in the studies included in each meta-analysis and we have highlighted where the reviews include or exclude studies and why this was done when we could determine a reason.

Please comment the quality of evidence and risk of bias of studies in the included systematic reviews. If quality of evidence is low, it is impossible to conclude. I suggest you analyse the quality of evidence of included reviews and change the conclusion with a more moderated statement.

We agree that if the studies included in the meta-analyses are weak then the conclusions may be weak. Nevertheless, the meta-analyses we have included have mostly been published in high impact journals with good statistical authors so we feel confident in the results they have presented. Nevertheless, we have moderated our conclusions to say that nut consumption appears to be associated with lower mortality and reduced CVD incidence

Reviewer 3 Report

This is a well written and comprehensive systematic review of meta-analyses on the association between nut consumption and cardio-metabolic diseases. 

 I have a few minor comments: 

 - It is difficult to quickly pull out information from summary Tables 1 and 2. It would be helpful to add two additional columns, one with population characteristics such as population size, age and follow-up period and another with the number of cohort studies included in the meta-analyses. It is also difficult to interpret what type of nuts were investigated in each study. A third column which clearly states this would be beneficial. 

-      There is no mention of the possible mechanisms of action. This is briefly mentioned in the introduction, but I think it would be useful to have a brief paragraph in the discussion detailing this. 

-      Also, do most of the studies look at total nut consumption? Is there a difference between types of nuts? I think this needs to be mentioned in the discussion. 

-      What future work needs to be done in this area? Is there sufficient evidence to support recommendations to increase nut intake for prevention of cardio-metabolic diseases? Please add this to the discussion/conclusion section. 

Author Response

Comments and Suggestions for Authors

This is a well written and comprehensive systematic review of meta-analyses on the association between nut consumption and cardio-metabolic diseases. 

I have a few minor comments: 

- It is difficult to quickly pull out information from summary Tables 1 and 2. It would be helpful to add two additional columns, one with population characteristics such as population size, age and follow-up period and another with the number of cohort studies included in the meta-analyses. It is also difficult to interpret what type of nuts were investigated in each study. A third column which clearly states this would be beneficial. 

Thank you. The table 1 and table 2 have been modified reflecting the suggestion.

-      There is no mention of the possible mechanisms of action. This is briefly mentioned in the introduction, but I think it would be useful

Thank you. The possible mechanisms of action have been addressed with a brief paragraph in the discussion detailing it.

“Nuts are comprised of unique macro and micronutrients (e.g. monounsaturated fatty acids, polyunsaturated fatty acids, fiber, arginine and magnesium) and bioactive compounds (e.g. phytosterols and polyphenols). The synergistic beneficial effects of nuts on cardiometabolic disease result from these nutrient- and non-nutrient compositions of nuts. The potential mechanisms underlying these effects can be explained through the improvement of glucose control, lipid profiles, weight control, endothelial function and gut microbiota modification [7,217,218]. “

 -      Also, do most of the studies look at total nut consumption? Is there a difference between types of nuts? I think this needs to be mentioned in the discussion. 

There appears to be little obvious differences with different nuts and we have now mentioned this in the discussion

-      What future work needs to be done in this area? Is there sufficient evidence to support recommendations to increase nut intake for prevention of cardio-metabolic diseases? Please add this to the discussion/conclusion section. 

 I think it is reasonably clear that nut consumption appears to be associated with lower rates of CVD and CVD mortality with clear mechanisms for this effect including lower lipids and improved endothelial function. What is required are more intervention studies to clarify the effects of nuts on glucose homeostasis and incidence of type 2 diabetes. Weight control also requires more work to demonstrate a direct effect. This has been addressed in the discussion.

Round  2

Reviewer 2 Report

I would like to thank the authors for this new version of the review and good consideration of previous suggestions. Most comments were taken in account which greatly improved the manuscript.

My only comment is in regard with the quality of evidence (lines 402-404). Quality of evidence in meta-analysis is usually assessed using the GRADE approach or any similar validated tool to evaluate the quality of results. Results from a meta-analysis composed of high-biased studies will be surely biased, independently of the number of subjects or the number of studies included in the meta-analysis. Further, the quality of a meta-analysis cannot be assumed by the journal’s impact factor. Therefore, the following sentence should be removed: The primary studies in the meta-analyses are in most cases high quality and large and have been published in high impact journals as have the meta-analyses so the overall conclusions are sound.

I suggest verifying if included meta-analyses evaluated the quality of evidence using GRADE or another validated tool and to discuss what were the results of these assessments. In my opinion, discussing potential biases in the results of the meta-analyses would increase the strength of your manuscript. 

Author Response

I would like to thank the authors for this new version of the review and good consideration of previous suggestions. Most comments were taken in account which greatly improved the manuscript.

My only comment is in regard with the quality of evidence (lines 402-404). Quality of evidence in meta-analysis is usually assessed using the GRADE approach or any similar validated tool to evaluate the quality of results. Results from a meta-analysis composed of high-biased studies will be surely biased, independently of the number of subjects or the number of studies included in the meta-analysis. Further, the quality of a meta-analysis cannot be assumed by the journal’s impact factor. Therefore, the following sentence should be removed: The primary studies in the meta-analyses are in most cases high quality and large and have been published in high impact journals as have the meta-analyses so the overall conclusions are sound.

Thank you. The sentence has been removed at lines 402-404.

I suggest verifying if included meta-analyses evaluated the quality of evidence using GRADE or another validated tool and to discuss what were the results of these assessments. In my opinion, discussing potential biases in the results of the meta-analyses would increase the strength of your manuscript.

We have added the following at line 402-405.

“All of the meta-analyses on total mortality and cause-specific mortality assessed the study quality using the Newcastle-Ottawa scale or a very similar scale and found no effect of study quality on the outcomes of the meta-analyses” 
